# Targeted genomic sequencing with probe capture for discovery and surveillance of coronaviruses in bats

Kevin S Kuchinski[1,2], Kara D Loos[3,4], Danae M Suchan[3,4], Jennifer N Russell[3,4], Ashton N Sies[3,4], Charles Kumakamba[5], Francisca Muyembe[5], Placide Mbala Kingebeni[5,6], Ipos Ngay Lukusa[5], Frida N'Kawa[5], Joseph Atibu Losoma[5], Maria Makuwa[5,7], Amethyst Gillis[8,9], Matthew LeBreton[10], James A Ayukekbong[11,12], Nicole A Lerminiaux[3,4], Corina Monagin[8,13], Damien O Joly[11,14], Karen Saylors[7,8], Nathan D Wolfe[8], Edward M Rubin[8], Jean J Muyembe Tamfum[6], Natalie A Prystajecky[1,2], David J McIver[11,15], Christian E Lange[7,11], Andrew DS Cameron[3,4]*

[1]Department of Pathology and Laboratory Medicine, University of British Columbia, Vancouver, Canada; [2]Public Health Laboratory, British Columbia Centre for Disease Control, Vancouver, Canada; [3]Department of Biology, Faculty of Science, University of Regina, Regina, Canada; [4]Institute for Microbial Systems and Society, Faculty of Science, University of Regina, Regina, Canada; [5]Metabiota Inc, Kinshasa, Democratic Republic of the Congo; [6]Institut National de Recherche Biomédicale, Kinshasa, Democratic Republic of the Congo; [7]Labyrinth Global Health Inc, St. Petersburg, United States; [8]Metabiota Inc, San Francisco, United States; [9]Development Alternatives, Washington, United States; [10]Mosaic, Yaoundé, Cameroon; [11]Metabiota, Nanaimo, Canada; [12]Southbridge Care, Cambridge, Canada; [13]One Health Institute, School of Veterinary Medicine, University of California, Davis, Davis, United States; [14]Nyati Health Consulting, Nanaimo, Canada; [15]Institute for Global Health Sciences, University of California, San Francisco, San Francisco, United States

*For correspondence:
andrew.cameron@uregina.ca

**Abstract** Public health emergencies like SARS, MERS, and COVID-19 have prioritized surveillance of zoonotic coronaviruses, resulting in extensive genomic characterization of coronavirus diversity in bats. Sequencing viral genomes directly from animal specimens remains a laboratory challenge, however, and most bat coronaviruses have been characterized solely by PCR amplification of small regions from the best-conserved gene. This has resulted in limited phylogenetic resolution and left viral genetic factors relevant to threat assessment undescribed. In this study, we evaluated whether a technique called hybridization probe capture can achieve more extensive genome recovery from surveillance specimens. Using a custom panel of 20,000 probes, we captured and sequenced coronavirus genomic material in 21 swab specimens collected from bats in the Democratic Republic of the Congo. For 15 of these specimens, probe capture recovered more genome sequence than had been previously generated with standard amplicon sequencing protocols, providing a median 6.1-fold improvement (ranging up to 69.1-fold). Probe capture data also identified five novel *alpha*- and *betacoronaviruses* in these specimens, and their full genomes were recovered with additional deep sequencing. Based on these experiences, we discuss how probe capture could be effectively operationalized alongside other sequencing technologies for high-throughput, genomics-based discovery and surveillance of bat coronaviruses.

## Editor's evaluation

This work applies hybrid-capture sequencing for coronavirus (CoV) surveillance in bats. Given that bats are a major reservoir for animal-to-human virus spillover events, which have caused several major epidemics/pandemics, this is a very important field of research. The reported hybrid-capture method shows some clear advantages over amplicon-based viral sequencing, which is the established standard in the field. This new approach has clear merits that are well supported by the data presented and is likely to become an important tool in viral surveillance programs that ultimately aim to predict/prevent/prepare for future pandemics. The work will be of interest to microbiologists, particularly those studying viruses or interested in genomics surveillance.

## Introduction

*Orthocoronavirinae*, commonly known as coronaviruses (CoVs), are a diverse subfamily of RNA viruses that infect a broad range of mammals and birds (*Corman et al., 2018*; *Ye et al., 2020*; *Ruiz-Aravena et al., 2021*). Since the 1960s, four endemic human CoVs have been identified as common causes of mild respiratory illnesses (*Corman et al., 2018*; *Ye et al., 2020*). In the past two decades, additional CoV threats have emerged, most notably SARS-CoV, MERS-CoV, and SARS-CoV-2, causing severe disease, public health emergencies, and global crises (*Drosten et al., 2003*; *Zaki et al., 2012*; *Hu et al., 2015*; *Corman et al., 2018*; *Ye et al., 2020*; *Zhou et al., 2020*). These spill-overs have established CoVs alongside influenza A viruses as important zoonotic pathogens and pandemic threats. Indeed, evolving perceptions of CoV risk have led to speculation that some historical pandemics have been mis-attributed to influenza, and they may have in fact been the spill-overs of now-endemic human CoVs (*Vijgen et al., 2005*; *Corman et al., 2018*; *Brüssow and Brüssow, 2021*).

Emerging CoV threats have motivated extensive viral discovery and surveillance activities at the interface between humans, livestock, and wildlife (*Drexler et al., 2014*; *Frutos et al., 2021*; *Geldenhuys et al., 2021*). Many of these activities have focused on bats (order *Chiroptera*). They are the second-most diverse order of mammals, following rodents, and they are a vast reservoir of CoV diversity (*Drexler et al., 2014*; *Hu et al., 2015*; *Frutos et al., 2021*; *Geldenhuys et al., 2021*; *Ruiz-Aravena et al., 2021*). Bats have been implicated in the emergence of SARS-CoV, MERS-CoV, SARS-CoV-2, and, less recently, the endemic human CoVs NL63 and 229E (*Li et al., 2005*; *Pfefferle et al., 2009*; *Tong et al., 2009*; *Huynh et al., 2012*; *Corman et al., 2015*; *Hu et al., 2015*; *Yang et al., 2015*; *Tao et al., 2017*; *Ye et al., 2020*; *Zhou et al., 2020*; *Ruiz-Aravena et al., 2021*).

Genomic sequencing has been instrumental for characterizing CoV diversity and potential zoonotic threats, but recovering viral genomes directly from animal specimens remains a laboratory challenge. Host tissues and microbiota contribute excessive background genomic material to specimens, diluting viral genome fragments and vastly increasing the sequencing depth required for target detection and accurate genotyping. Consequently, laboratory methods for targeted enrichment of viral genome material have been necessary for practical, high-throughput sequencing of surveillance specimens (*Houldcroft et al., 2017*; *Fitzpatrick et al., 2021*).

There are two major paradigms for targeted enrichment of genomic material. The first, called amplicon sequencing, uses PCR to amplify target genomic material. It is comparatively straightforward and sensitive, but PCR chemistry limits amplicon length and relies on the presence of specific primer sites across diverse taxa (*Houldcroft et al., 2017*; *Fitzpatrick et al., 2021*). In practice, extensive genomic divergence within viral taxa often constrains amplicon locations to the most conserved genes, limiting phylogenetic resolution (*Drexler et al., 2014*; *Li et al., 2020*). This also hinders characterization of viral genetic factors relevant for threat assessment like those encoding determinants of host range, tissue tropism, and virulence. These kinds of targets are often hypervariable due to strong evolutionary pressures from host adaptation and immune evasion, and consequently they do not have well-conserved locations for PCR primers. Due to these limitations, studies of CoV diversity have been almost exclusively based on small regions of the relatively conserved RNA-dependent RNA polymerase (RdRp) gene (*Drexler et al., 2014*; *Geldenhuys et al., 2021*).

The second major paradigm for enriching viral genomic material is called hybridization probe capture. This method uses longer nucleotide oligomers to anneal and immobilize complementary target genomic fragments while background material is washed away. Probes are typically 80–120 nucleotides in length, making them more tolerant of sequence divergence and nucleotide mismatches

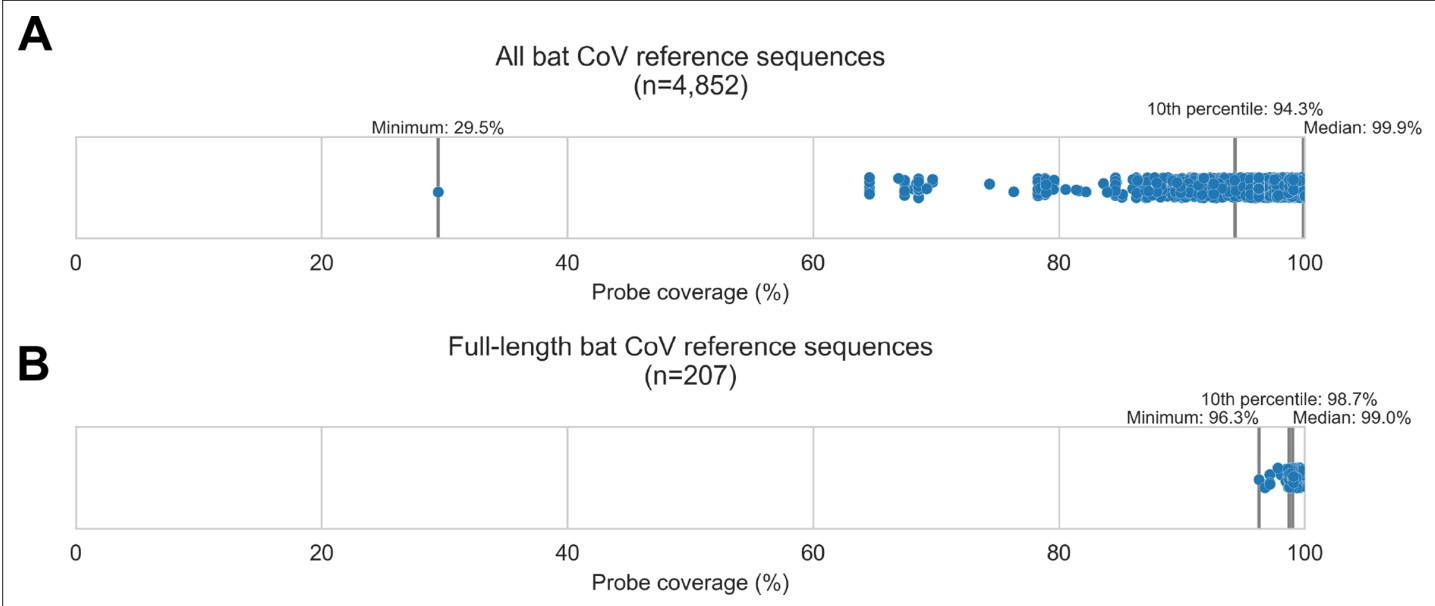

**Figure 1.** Custom hybridization probe panel provided broadly inclusive coverage of known bat coronavirus diversity in silico. Bat coronavirus (CoV) sequences were obtained by downloading all available *alphacoronavirus*, *betacoronavirus*, and unclassified *coronaviridae* and *coronavirinae* sequences from GenBank on 4 October 2020 and searching for bat-related keywords in sequence headers. A custom panel of 20,000 probes was designed to target these sequences using the *makeprobes* module in the ProbeTools package. The ProbeTools *capture* and *stats* modules were used to assess probe coverage of bat CoV reference sequences. (**A**) Each bat CoV sequence is represented as a dot plotted according to its probe coverage, that is, the percentage of its nucleotide positions covered by at least one probe in the custom panel. (**B**) The same analysis was performed on the subset of sequences representing full-length genomes (>25 kb in length).

than PCR primers (*Brown et al., 2016*). Probe panels are also highly scalable, allowing for the simultaneous capture of thousands to millions of target sequences. This has made them popular for applications where diverse and hypervariable viruses are targeted (*Bonsall et al., 2015*; *Briese et al., 2015*; *O'Flaherty et al., 2018*; *Wylezich et al., 2021*; *Wylie et al., 2015*). Probe capture has only been occasionally used to attempt sequencing of bat CoVs, however (*Lim et al., 2019*; *Li et al., 2020*).

In this study, we evaluated hybridization probe capture for enriching CoV genomic material in oral and rectal swabs previously collected from bats. We designed a custom panel of 20,000 hybridization probes targeting the known diversity of bat CoVs. This panel was applied to 21 swab specimens collected in the Democratic Republic of the Congo (DRC), in which novel CoVs had been previously characterized by partial RdRp sequencing using standard amplicon methods (*Kumakamba et al., 2021*). We compared the extent of genome recovery by probe capture and amplicon sequencing, and we used probe capture data in conjunction with deep metagenomic sequencing to characterize full genomes for five novel *alpha-* and *betacoronaviruses*. Based on these experiences, we discuss how probe capture could be effectively operationalized alongside other targeted sequencing technologies for high-throughput, genomics-based discovery and surveillance of bat CoVs.

## Results

### Custom hybridization probe panel provided broad coverage in silico of known bat CoV diversity

To begin this study, we designed a custom panel of hybridization probes targeting known bat CoV diversity. We obtained 4,852 bat CoV genomic sequences from GenBank, used them to design a custom panel of 20,000 probe sequences, then assessed in silico how extensively these reference sequences were covered by our custom panel (*Figure 1A*). For 90% of these bat CoV sequences, the custom panel covered at least 94.32% of nucleotide positions. We also evaluated probe coverage for the subset of these sequences representing full-length bat CoV genomes (*Figure 1B*), and 90% of

these targets had at least 98.73% of their nucleotide positions covered. These results showed broad probe coverage of known bat CoV diversity at the time the panel was designed.

## Probe capture provided more extensive genome recovery than previous amplicon sequencing for most specimens

We used our custom panel to assess probe capture recovery of CoV material in 25 metagenomic sequencing libraries. We prepared these libraries from a retrospective collection of 21 bat oral and rectal swabs that had been collected in DRC between 2015 and 2018 (*Kumakamba et al., 2021*). These swabs had been collected as part of the PREDICT project, a large-scale United States Agency for International Development (USAID) Emerging Pandemic Threats initiative that has collected over 20,000 animal specimens from 20 CoV hotspot countries (e.g. *Anthony et al., 2017*; *Lacroix et al., 2017*; *Nziza et al., 2020*; *Valitutto et al., 2020*; *Ntumvi et al., 2022*). Most libraries (n=19) were prepared from archived RNA that had been previously extracted from these specimens, although

**Table 1.** Bat specimens and sequencing libraries analysed in this study.

Rectal and oral swabs collected for a previous study were used to evaluate hybridization probe capture (*Kumakamba et al., 2021*). For 19 swabs, archived RNA extracted during the previous study was assayed. For 6 swabs, freshly extracted RNA (using a conventional Trizol method) was assayed. As part of the previous study, Kumakamba et al. generated partial sequences from the RNA-dependent RNA polymerase gene, which were used to assign *alpha*- and *betacoronaviruses* in these specimens to four novel phylogenetic groups.

| Specimen ID | Library ID | Host | Swab type | RNA extraction method | Phylogenetic group |
|---|---|---|---|---|---|
| CDAB0017RSV | CDAB0017RSV-PRE | *Micropteropus pusillus* | Rectal | Previously extracted | W-Beta-2 |
| CDAB0040R | CDAB0040R-PRE | *Myonycteris sp.* | Rectal | Previously extracted | W-Beta-2 |
| CDAB0040RSV | CDAB0040RSV-PRE | *Myonycteris sp.* | Rectal | Previously extracted | W-Beta-2 |
| CDAB0305R | CDAB0305R-PRE | *Micropteropus pusillus* | Rectal | Previously extracted | W-Beta-2 |
| CDAB0146R | CDAB0146R-PRE | *Eidolon helvum* | Rectal | Previously extracted | W-Beta-3 |
| CDAB0158R | CDAB0158R-PRE | *Eidolon helvum* | Rectal | Previously extracted | W-Beta-3 |
| CDAB0160R | CDAB0160R-PRE | *Eidolon helvum* | Rectal | Previously extracted | W-Beta-3 |
| CDAB0173R | CDAB0173R-PRE | *Eidolon helvum* | Rectal | Previously extracted | W-Beta-3 |
| CDAB0174R | CDAB0174R-PRE | *Eidolon helvum* | Rectal | Previously extracted | W-Beta-3 |
| CDAB0203R | CDAB0203R-PRE | *Eidolon helvum* | Rectal | Previously extracted | W-Beta-3 |
| CDAB0212R | CDAB0212R-PRE | *Eidolon helvum* | Rectal | Previously extracted | W-Beta-3 |
| CDAB0217R | CDAB0217R-PRE | *Eidolon helvum* | Rectal | Previously extracted | W-Beta-3 |
| CDAB0113RSV | CDAB0113RSV-PRE | *Hipposideros cf. ruber* | Rectal | Previously extracted | W-Beta-4 |
| CDAB0486R | CDAB0486R-PRE | *Chaerephon sp.* | Rectal | Previously extracted | Q-Alpha-4 |
| CDAB0488R | CDAB0488R-PRE | *Mops condylurus* | Rectal | Previously extracted | Q-Alpha-4 |
| CDAB0488R | CDAB0488R-TRI | *Mops condylurus* | Rectal | Trizol re-extraction | Q-Alpha-4 |
| CDAB0491R | CDAB0491R-PRE | *Mops condylurus* | Rectal | Previously extracted | Q-Alpha-4 |
| CDAB0491R | CDAB0491R-TRI | *Mops condylurus* | Rectal | Trizol re-extraction | Q-Alpha-4 |
| CDAB0492R | CDAB0492R-PRE | *Mops condylurus* | Rectal | Previously extracted | Q-Alpha-4 |
| CDAB0492R | CDAB0492R-TRI | *Mops condylurus* | Rectal | Trizol re-extraction | Q-Alpha-4 |
| CDAB0494O | CDAB0494O-TRI | *Mops condylurus* | Oral | Trizol re-extraction | Q-Alpha-4 |
| CDAB0494R | CDAB0494R-PRE | *Mops condylurus* | Rectal | Previously extracted | Q-Alpha-4 |
| CDAB0494R | CDAB0494R-TRI | *Mops condylurus* | Rectal | Trizol re-extraction | Q-Alpha-4 |
| CDAB0495O | CDAB0495O-PRE | *Mops condylurus* | Oral | Previously extracted | Q-Alpha-4 |
| CDAB0495R | CDAB0495R-TRI | *Mops condylurus* | Rectal | Trizol re-extraction | Q-Alpha-4 |

**Table 2.** Sequencing metrics for probe captured libraries.

Total reads and sequencing output were measured for each library. Raw metrics describe unprocessed FASTQ files directly from the sequencer. Valid metrics describe FASTQ files following pre-processing to trim adapters, trim trailing low-quality bases, remove index hops, and remove PCR chimeras. On-target metrics were estimated by mapping valid data to the coronavirus reference sequence selected for each specimen and to the contigs assembled from each specimen.

| Library ID | Raw reads (#) | Raw output (kb) | Valid reads (#) | Valid output (kb) | Mapped reads (#) | Mapped size (kb) |
|---|---|---|---|---|---|---|
| CDAB0017RSV-PRE | 115,280 | 14,225.8 | 47,609 | 7362 | 36,716 | 4919 |
| CDAB0040R-PRE | 37,950 | 4708.9 | 695 | 121.1 | 0 | 0 |
| CDAB0040RSV-PRE | 373,254 | 45,333.7 | 176,783 | 26,783.3 | 48,136 | 7068.4 |
| CDAB0113RSV-PRE | 31,394 | 4261 | 16,861 | 2875 | 16,302 | 2772.2 |
| CDAB0146R-PRE | 11,870 | 1520 | 193 | 23.7 | 186 | 22.6 |
| CDAB0158R-PRE | 48,014 | 6422.5 | 4189 | 706.9 | 1548 | 239.8 |
| CDAB0160R-PRE | 83,524 | 9948.5 | 2513 | 376.6 | 1363 | 191.2 |
| CDAB0173R-PRE | 10,628 | 1403 | 206 | 34.4 | 203 | 33.7 |
| CDAB0174R-PRE | 900,578 | 118,821 | 107,979 | 17,525.4 | 82,679 | 13,290.9 |
| CDAB0203R-PRE | 6,832,218 | 849,284.9 | 1,186,186 | 188,188.9 | 456,474 | 68,384.7 |
| CDAB0212R-PRE | 60,838 | 7526 | 4158 | 681.4 | 4152 | 678.3 |
| CDAB0217R-PRE | 20,078,142 | 2,617,427.3 | 8,946,935 | 1,467,955.3 | 5,173,448 | 81,5381.3 |
| CDAB0305R-PRE | 27,054 | 3182.7 | 3971 | 594.9 | 1787 | 250 |
| CDAB0486R-PRE | 442,456 | 5,8326.7 | 56,838 | 9377 | 20,687 | 3385.1 |
| CDAB0488R-PRE | 188,294 | 24,679 | 2913 | 506.2 | 2867 | 493.1 |
| CDAB0488R-TRI | 343,916 | 45,867.1 | 8415 | 1381.2 | 8225 | 1346.2 |
| CDAB0491R-PRE | 791,120 | 96,136.3 | 46,509 | 7081.8 | 45,289 | 6854.6 |
| CDAB0491R-TRI | 1,561,144 | 204,995.4 | 173,533 | 29,280.5 | 157,889 | 26,421.2 |
| CDAB0492R-PRE | 3,453,456 | 448,217.9 | 277,176 | 48,023.8 | 185,665 | 31,641.3 |
| CDAB0492R-TRI | 4,200,520 | 518,837.7 | 139,804 | 20,074.7 | 93,442 | 13,294.3 |
| CDAB0494O-TRI | 141,494 | 18,980.9 | 290 | 49.7 | 60 | 11.3 |
| CDAB0494R-PRE | 82,360 | 11,162.7 | 22 | 4 | 0 | 0 |
| CDAB0494R-TRI | 95,924 | 12,762.1 | 9 | 2.1 | 0 | 0 |
| CDAB0495O-PRE | 27,074 | 3776 | 0 | 0 | 0 | 0 |
| CDAB0495R-TRI | 470,850 | 63,267 | 8896 | 1440.9 | 8672 | 1399.2 |

some libraries (n=6) were prepared from RNA that was freshly extracted from archived primary specimens (*Table 1*). CoVs had been previously detected in these specimens with PCR assays by *Quan et al., 2010*, and *Watanabe et al., 2010*. Sanger sequencing of these amplicons by *Kumakamba et al., 2021*, had generated partial RdRp sequences of 286 or 387 nucleotides, which had been used to assign these specimens to four novel phylogenetic groups of *alpha-* and *betacoronaviruses* (*Table 1*).

We captured CoV genomic material in these metagenomic bat swab libraries with our custom probe panel then performed genomic sequencing (*Table 2*). To assess CoV recovery, we began with a strategy that would be suitable for automated bioinformatic analysis in high-throughput surveillance settings: sequencing reads from probe captured libraries were assembled de novo into contigs, then CoV sequences were identified by locally aligning contigs against a database of CoV reference sequences. In total, 113 CoV contigs were recovered from 17 of 25 libraries. We compared contig lengths to the partial RdRp amplicons that been previously generated for these specimens (*Figure 2A*).

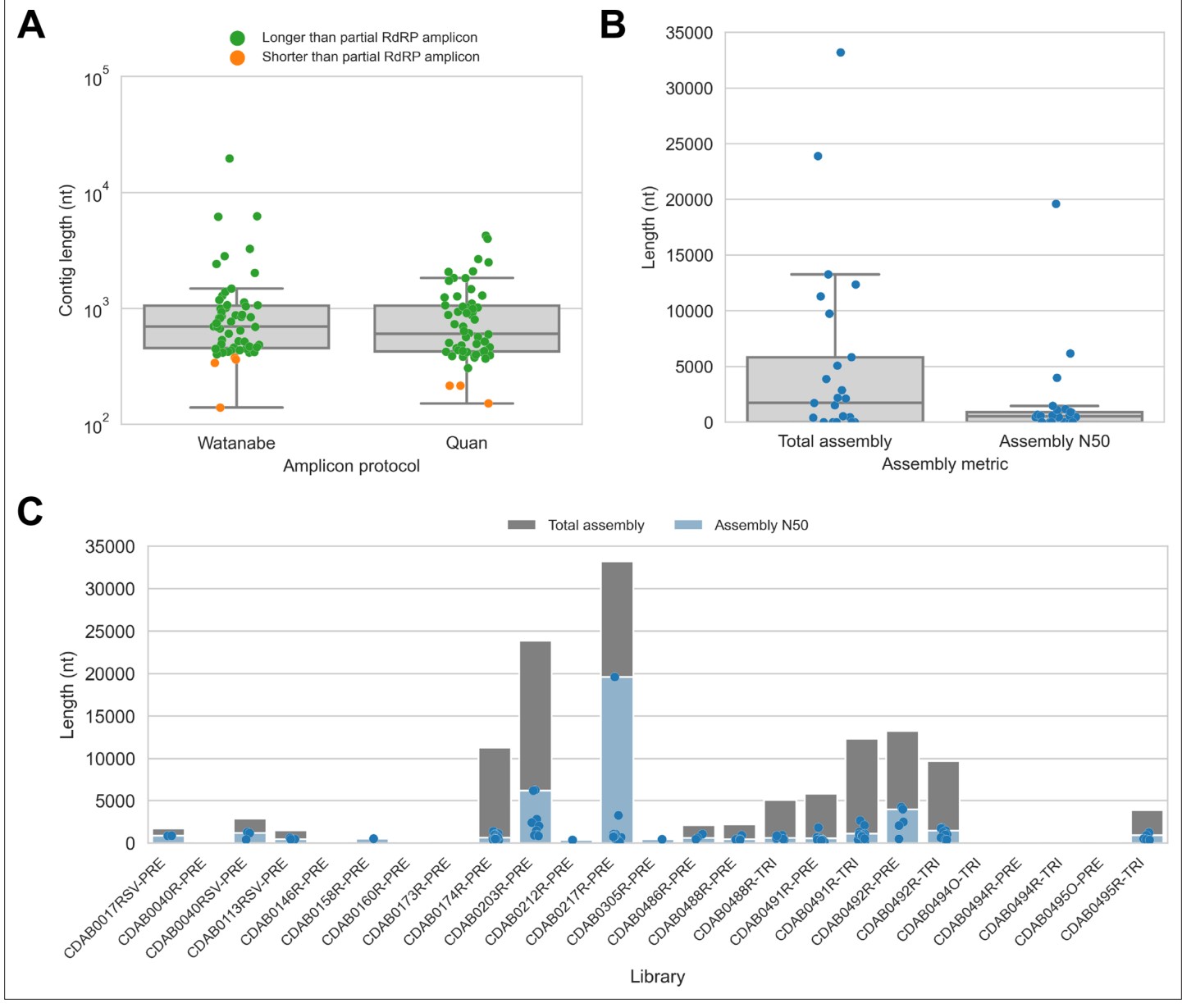

**Figure 2.** De novo assembly of probe captured libraries yielded more genome sequence than standard amplicon sequencing methods for most specimens. Reads from probe captured libraries were assembled de novo with coronaSPAdes, and coronavirus contigs were identified by local alignment against a database of all *coronaviridae* sequences in GenBank. (**A**) The size distribution of contigs from all libraries is shown. Dots are coloured to indicate whether the length of the contig exceeded partial RNA-dependent RNA polymerase (RdRP) gene amplicons previously sequenced from these specimens. (**B**) Total assembly size and assembly N50 distributions for all libraries. (**C**) Each contig is represented as a dot plotted according to its length. Assembly N50 sizes and total assembly sizes are indicated by the height of their bars.

The protocol by Watanabe et al. had generated 387 nucleotide-long partial RdRp sequences, but median contig size with probe capture for these specimens was 696 nucleotides (IQR: 453–1051 nucleotides, max: 19,601 nucleotides). The protocol by Quan et al. had generated 286 nucleotide-long partial RdRp sequences, but median contig size with probe capture for these specimens was 602 nucleotides (IQR: 423–1053 nucleotides, max: 4240 nucleotides). Overall, 107 contigs (93.8%) were longer than the partial RdRp sequence previously generated for their specimen by standard amplicon sequencing protocols, demonstrating the capacity of probe capture to recover larger contiguous fragments of CoV genome sequence. We also assessed nucleotide sequence concordance; for specimens where the partial RdRp amplicon sequence was successfully assembled, nucleotide identities ranged from 99.3% to 100% (median = 100%, maximum two mismatches).

Next, we used assembly size metrics to assess the extent to which these contigs represented complete genomes. The median total assembly size was 1724 nucleotides (IQR: 0–5834 nucleotides), while median assembly N50 size was 533 nucleotides (IQR: 0–908 nucleotides) (*Figure 2B*). This assembly size-based assessment of genome completeness had limitations, however. Some assembly sizes may have been understated by genome regions with comparatively low read coverage that failed to assemble. Conversely, other assembly sizes may have been overstated by redundant contigs resulting from forked assembly graphs, either due to genetic variation within the intrahost viral population or due to polymerase errors introduced during library construction and probe capture. For instance, the total assembly size for library CDAB0217R-PRE was 33,195 nucleotides, exceeding the length of the longest known CoV genome (*Figure 2C*). Another limitation of this analysis was that these assembly metrics provided no indication of which regions of the genome had been recovered.

To address these limitations, we also applied a reference sequence-based strategy. We used the contigs to identify the best available CoV reference sequences for each of the four novel phylogenetic groups to which these specimens had been assigned. Sequencing reads from captured libraries were directly mapped to these reference sequences and the contigs we had assembled de novo were also locally aligned to them (*Figure 3* and *Figure 3—figure supplements 1–4*). Based on these read mappings and contig alignments, we calculated for each library a breadth of reference sequence recovery, that is, the number of nucleotide positions in the reference sequence covered by either mapped sequencing reads or contigs (*Figure 4A*). The number of reads mapped to these reference sequences and contigs was also used to estimate on-target rates for these libraries (*Table 2*).

The median breadth of reference sequence recovery for all libraries was 2376 nucleotides (IQR: 306–9446 nucleotides). Most libraries (48%) represented specimens from phylogenetic group Q-Alpha-4, which had a median reference sequence recovery of 6497 nucleotides (IQR: 733–9802 nucleotides, max: 12,673 nucleotides). Phylogenetic group W-Beta-3 also accounted for a substantial fraction of libraries (32%), and although median reference sequence recovery was lower than for Q-Alpha-4 (2427 nucleotides), W-Beta-3 provided the libraries with the most extensive reference sequence recoveries (IQR: 780–19,286 nucleotides, max: 26,755 nucleotides). As a simple way to quantify differences in recovery of CoV genome sequence between probe capture and amplicon sequencing, we calculated the ratio between the breadth of reference sequence recovery and the length of the previously generated partial RdRp amplicon sequence for each library (*Figure 4B*). The median ratio was 6.1-fold (IQR: 0.8-fold to 33.0-fold), reaching a maximum of 69.1-fold. Probe capture recovery was greater for 18 of 25 libraries (72%), representing 15 of 21 specimens (71%).

We also used reference sequence coverage to estimate the completeness of recovery for the RdRp and spike genes (*Figure 4C*). Overall, RdRp was more completely recovered than spike. Furthermore, following the overall extent of recovery trend observed in *Figure 4A*, recovery of RdRp and spike was more complete for viruses from phylogenetic group Q-Alpha-4 than the *betacoronavirus* groups, although multiple complete RdRp genes were recovered from both Q-Alpha-4 and W-Beta-3 groups. No complete spike genes were recovered.

## Probe capture recovery limited by in vitro sensitivity

No CoV sequences were recovered from 4 of 25 libraries (representing three specimens), despite partial RdRp sequences being obtained from them previously. Furthermore, probe capture did not yield any complete CoV genomes, and many specimens displayed scattered and discontinuous reference sequence coverage (*Figure 3—figure supplements 1–4*). We considered two explanations for this result. First, CoV material in these libraries may not have been completely captured because they were not targeted by any probe sequences in the panel. Second, CoV material in these specimens may not have been incorporated into the sequencing libraries due to factors limiting in vitro sensitivity, for example, low prevalence of viral genomic material; suboptimal nucleic acid concentration and integrity in archived RNA and primary specimens; and library preparation reaction inefficiencies.

First, we assessed in vitro sensitivity. To exclude missing probe coverage as a confounder in this analysis, we evaluated recovery of the previously sequenced partial RdRp amplicons. Since their sequences were known, we could assess probe coverage in silico and demonstrate whether these targets were covered by the panel. All partial RdRp amplicons had at least 95.3% of their nucleotide positions covered by the probe panel (*Figure 5A*), but this did not translate into extensive recovery. For 12 of 25 libraries, no part of the partial RdRp sequence was recovered, and full/nearly

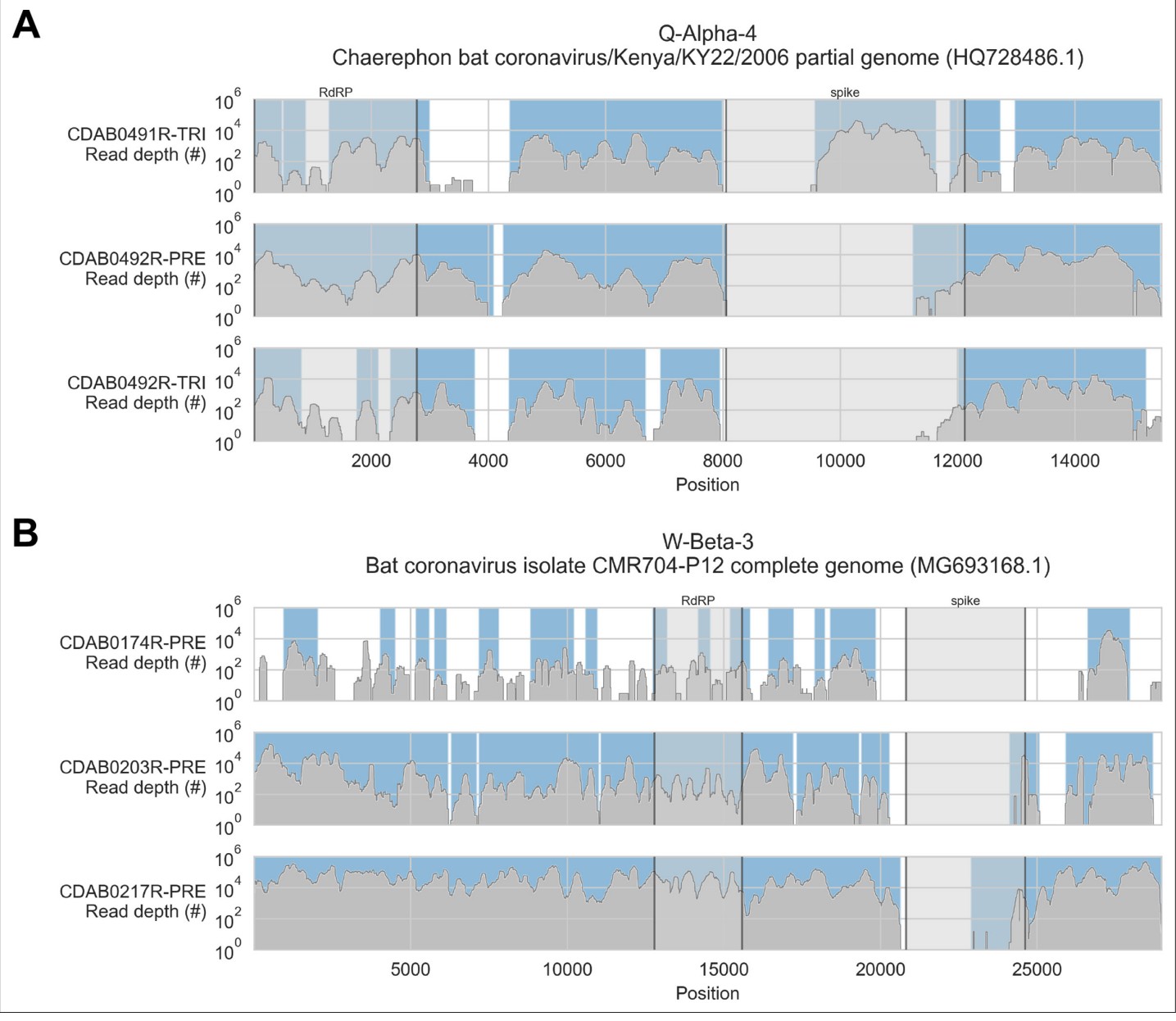

**Figure 3.** Coverage of reference sequences by probe captured libraries was used to assess extent and location of recovery. Reference sequences were chosen for each previously identified phylogenetic group (indicated in panel titles). Coverage of these reference sequences was determined by mapping reads and aligning contigs from probe captured libraries. Dark grey profiles show depth of read coverage along reference sequences. Blue shading indicates spans where contigs aligned. The locations of spike and RNA-dependent RNA polymerase (RdRP) genes are indicated in each reference sequence and shaded light grey. This figure shows the six libraries with the most extensive reference sequence coverage. Similar plots are provided as figure supplements for all libraries where any coronavirus sequence was recovered (*Figure 3—figure supplements 1–4*) .

The online version of this article includes the following figure supplement(s) for figure 3:

**Figure supplement 1.** Coverage of reference sequence by probe captured libraries for specimens from phylogenetic group Q-Alpha-4.

**Figure supplement 2.** Coverage of reference sequence by probe captured libraries for specimens from phylogenetic group W-Beta-2.

**Figure supplement 3.** Coverage of reference sequence by probe captured libraries for specimens from phylogenetic group W-Beta-3.

**Figure supplement 4.** Coverage of reference sequence by probe captured libraries for specimens from phylogenetic group W-Beta-4.

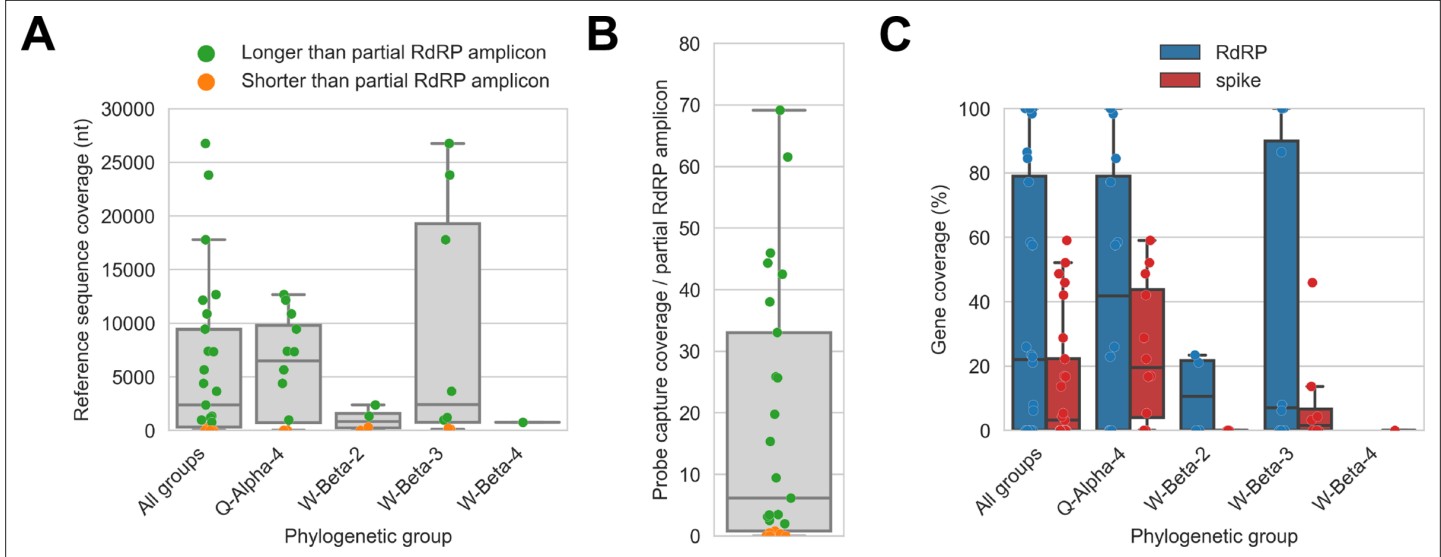

**Figure 4.** Probe captured libraries provided more extensive coverage of reference genomes than standard amplicon sequencing protocols for most specimens. Reference sequences were selected for the previously identified phylogenetic groups to which these specimens had been assigned by *Kumakamba et al., 2021*. (**A**) Coverage of these reference sequences was determined by mapping reads and aligning contigs from probe captured libraries. Each library is represented as a dot, and dots are coloured according to whether reference sequence coverage exceeded the length of the partial RNA-dependent RNA polymerase (RdRP) gene sequence that had been previously generated by amplicon sequencing. (**B**) The number of reference sequence positions covered by probe captured libraries was divided by the length of the partial RdRP amplicon sequences from these specimens. This provided the fold-difference in recovery between probe capture and standard amplicon sequencing methods. (**C**) Percent coverage of the spike and RdRP genes were calculated for each specimen.

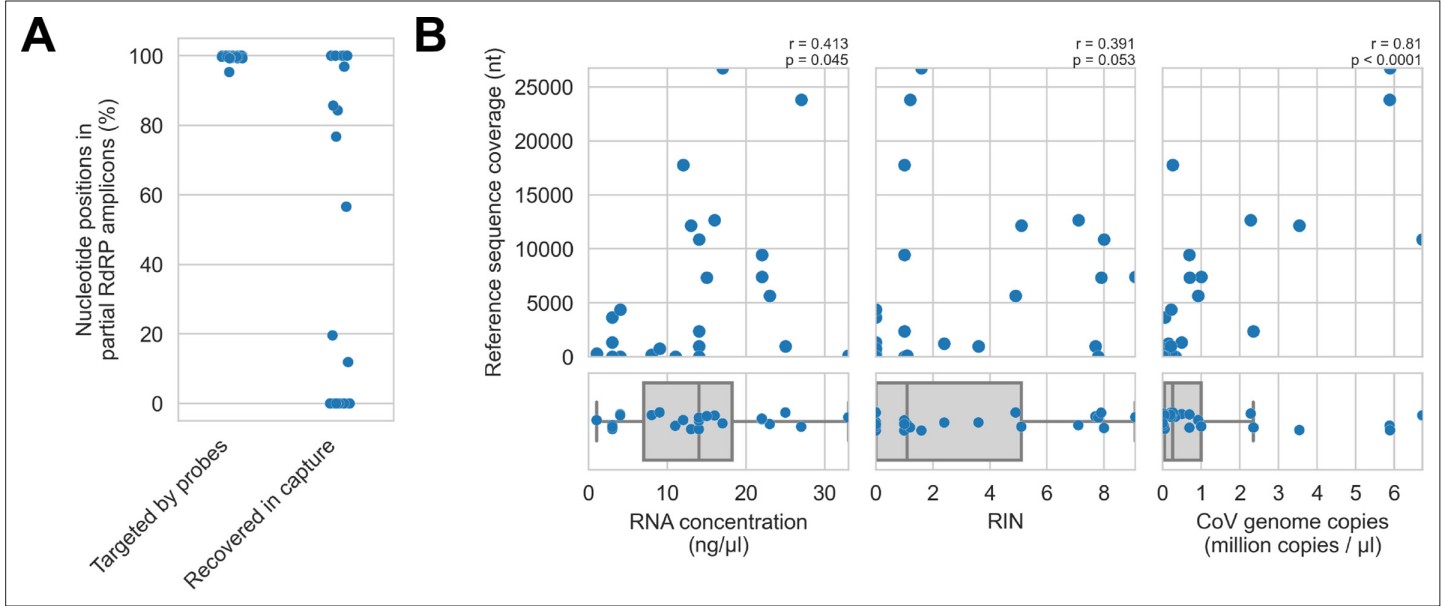

**Figure 5.** Recovery of coronavirus (CoV) genomic material was limited in vitro by method sensitivity. (**A**) Sensitivity was assessed by evaluating recovery of partial RNA-dependent RNA polymerase (RdRp) gene regions that had been previously sequenced in these specimens by amplicon sequencing. Probe coverage of partial RdRp sequences was assessed in silico to exclude insufficient probe design as an alternate explanation for incomplete recovery of these targets. (**B**) Input RNA concentration, RNA integrity numbers (RINs), and CoV genome abundance were measured for each specimen. The impact of these specimen characteristics on recovery by probe capture (as measured by reference sequence coverage) was assessed using Spearman's rank correlation (test results stated in plots). An outlier was omitted from this analysis: RNA concentration for specimen CDAB0160R was recorded as 190 ng/µl, a value 4.7 SDs from the mean of the distribution.

full recovery (>95%) of the partial RdRp sequence was achieved for only 7 of 25 libraries (*Figure 5A*). These results demonstrated that genome recovery had been limited by factors other than probe panel inclusivity.

Next, we examined nucleic acid concentration and integrity, two specimen characteristics associated with successful library preparation. Median RNA integrity number (RIN) values and RNA concentrations for these specimens were low: 1.1 and 14 ng/µl respectively, as was expected from archived material (*Figure 5B*). To assess the impact of RIN and RNA concentration on probe capture recovery, we compared these specimen characteristics against breadth of reference sequence recovery from the corresponding libraries (*Figure 5B*). Weak monotonic relationships were observed, with lower RNA concentration and lower RIN values generally leading to worse genome recovery. This relationship was significant for RNA concentration (p=0.045, Spearman's rank correlation), but not for RNA integrity despite trending towards significance (p=0.053, Spearman's rank correlation). These weak associations suggested additional factors hindered recovery, for example, low prevalence of viral material or missing probe coverage for genomic regions outside the partial RdRp target.

Using the previously generated partial RdRp sequences, we designed RT-qPCR assays to estimate CoV genome copies in these specimens (*Figure 5B*). The median abundance of viral material was 0.26 million genome copies/µl. There was a strong and significant monotonic relationship between viral abundance and extent of genome recovery (p<0.0001, Spearman's rank correlation).

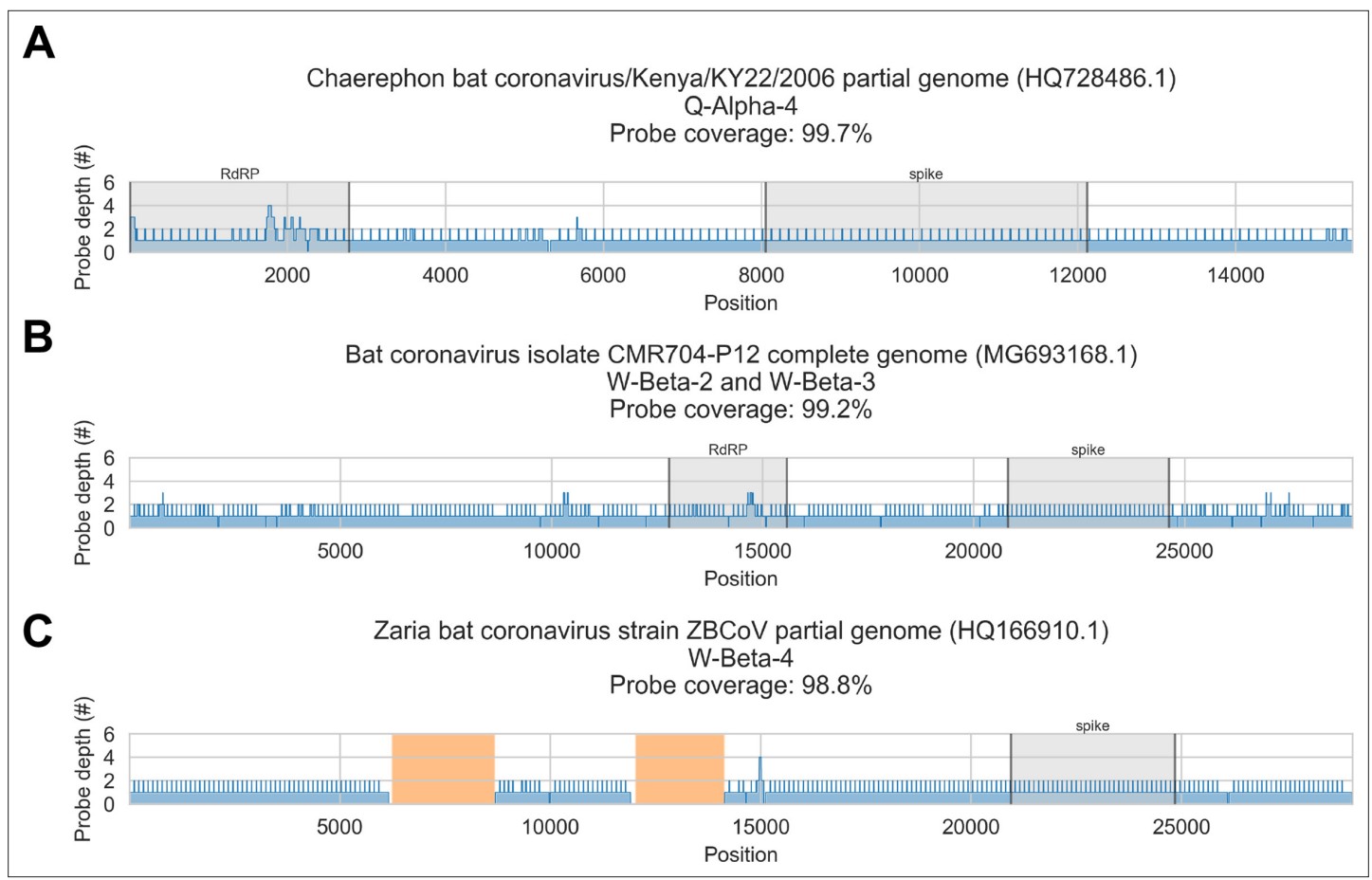

**Figure 6.** In silico assessment of probe panel coverage for reference genomes. Reference sequences were chosen for each previously identified phylogenetic group (indicated in panel titles). Blue profiles show the number of probes covering each nucleotide position along the reference sequence. Probe coverage, that is, the percentage of nucleotide positions covered by at least one probe, is stated in panel titles. Ambiguity nucleotides (Ns) are shaded in orange, and these positions were excluded from the probe coverage calculations. The locations of spike and RNA-dependent RNA polymerase (RdRP) genes are indicated in each reference sequence (where available) and shaded grey.

## Inclusivity of custom probe panel against CoV taxa in study specimens

Next, we considered if blind spots in the probe panel had contributed to incomplete genome recovery from these specimens. This inquiry suffered a counterfactual problem: to assess whether the CoV taxa in our specimens were fully covered by our probe panel, we would need their complete genome sequences. We did not have their full genome sequences, however, because the probes did not recover them. Instead, we evaluated probe coverage of the reference sequences assigned to each phylogenetic group, assuming they were the available CoV sequences most similar to those in our specimens.

Probe coverage was nearly complete for all reference sequences (*Figure 6*). Nonetheless, reference sequence recovery did not exceed 92.3% for any of these libraries, and complete spike genes were conspicuously absent (*Figure 3*, *Figure 3—figure supplement 1*, *Figure 3—figure supplement 2*, *Figure 3—figure supplement 3*, *Figure 3—figure supplement 4*). This included specimens like CDAB0203R-PRE, CDAB0217R-PRE, and CDAB0492R-PRE where recovery was otherwise extensive and contiguous, suggesting genomic material was sufficiently abundant and intact for sensitive library construction. These results indicated the presence of CoVs similar to bat CoV CMR704-P12 and *Chaerephon* bat corornavirus/Kenya/KY22/2006, except with novel spike genes that diverged from the spike genes of these reference sequences and all other CoVs described in GenBank.

## Recovery of complete genome sequences from five novel bat *alpha*- and *betacoronaviruses*

Analysis of our probe capture data confirmed the presence of several novel CoVs in these specimens, as had been previously determined by *Kumakamba et al., 2021*. Our results also suggested the CoVs in these specimens contained spike genes that were highly divergent from any others that have been previously described. This led us to perform deep metagenomic sequencing on select specimens to attempt recovery of complete CoV genomes. We selected the following nine specimens, either due to extensive recovery by probe capture (indicating comparatively abundant and intact viral genomic material) or to ensure representation of the four novel phylogenetic groups: CDAB0017RSV, CDAB0040RSV, CDAB0174R, CDAB0203R, CDAB0217R, CDAB0113RSV, CDAB0491R, and CDAB0492R.

Complete genomes were only recovered from five specimens: CDAB0017RSV, CDAB0040RSV, CDAB0203R, CDAB0217R, and CDAB0492R. The abundance of CoV genomic material in these five

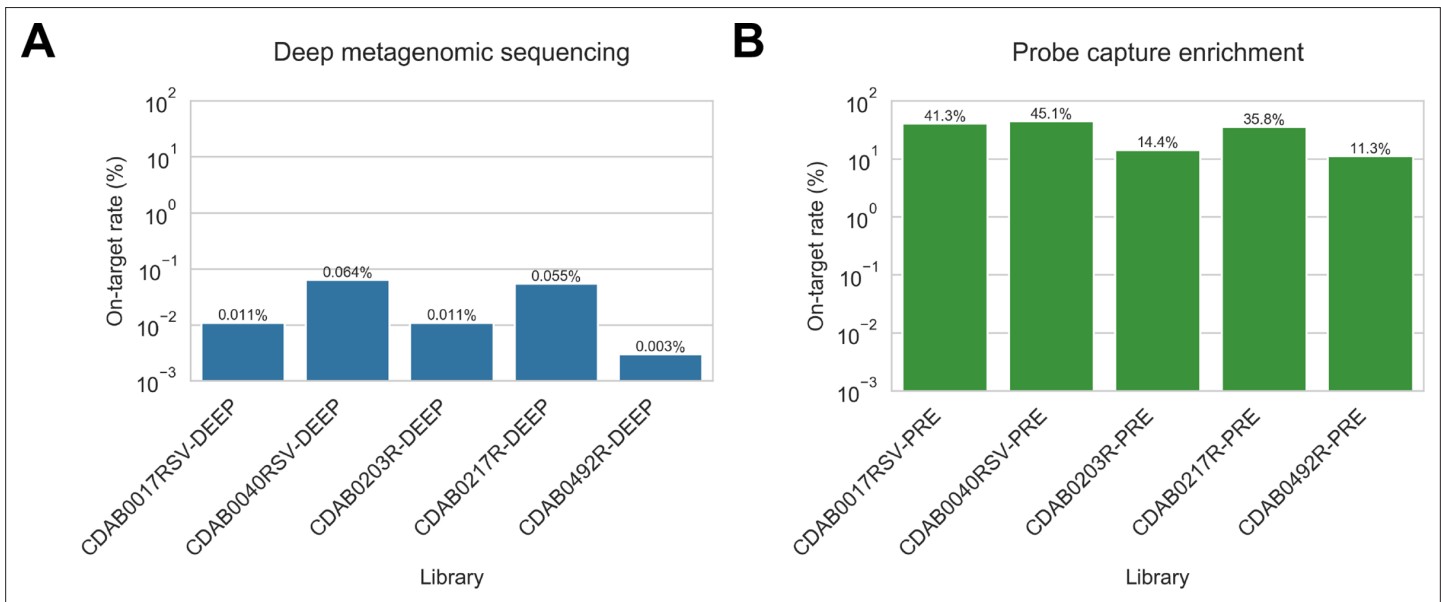

**Figure 7.** Coronavirus (CoV) genomic material was low abundance in swab specimens but effectively enriched by probe capture. (**A**) Reads from uncaptured, deep metagenomic sequenced libraries were mapped to complete genomes recovered from these specimens to assess abundance of CoV genomic material. On-target rate was calculated as the percentage of total reads mapping that mapped to the CoV genome sequence. (**B**) Reads from probe captured libraries were also mapped to assess enrichment and removal of background material. Most libraries used for probe capture (-PRE and -TRI) had insufficient volume remaining for deep metagenomic sequencing, so new libraries were prepared (-DEEP) from the same specimens.

specimens was estimated by mapping reads from uncaptured libraries to the complete genome sequence that we recovered. On-target rates, that is, the percentage of total reads mapping to the CoV genome, were calculated (*Figure 7A*). These ranged from 0.003% to 0.064%, revealing the extremely low abundance of viral genomic material present in these swabs. Considering these were the most successful libraries, these results highlighted that low prevalence of viral genomic material is one challenging characteristic of swab specimens.

We also used the complete genome sequences that we recovered to assess how effectively probe capture enriched target genomic material in these specimens. Valid reads from probe captured libraries were mapped to the complete genomes from their corresponding specimens. On-target rates for captured libraries ranged from 11.3% to 45.1% of valid reads (*Figure 7B*).

Due to insufficient library material remaining after probe capture, new libraries had been made for deep metagenomic sequencing. Consequently, we did not pair on-target rates for these libraries to calculate fold-enrichment values. Instead, we compared mean on-target rates for the deep-sequenced

**Figure 8.** Phylogenetic tree of translated spike gene sequences from *alphacoronaviruses*. Spike sequences are coloured according to whether they were from study specimens (blue), human CoVs (red), RefSeq (black), or GenBank (grey). Only the 25 closest-matching spike sequences from GenBank were included, as determined by blastp bitscores. GenBank and RefSeq accession numbers are provided in parentheses. The scale bar measures amino acid substitutions per site.

unenriched metagenomic libraries (0.029% mean on-target) against the original probe captured libraries (29.6% mean on-target); we observed a 1020-fold difference between these means, with the probe captured on-target rates significantly higher (p<0.001, t-test on two independent means). These results confirmed effective enrichment by probe capture of CoV material present in these libraries.

## Phylogenetic analysis of novel spike gene sequences

Novel spike gene sequences were translated from the complete genomes we had recovered, then these were compared to spike protein sequences from other CoVs in GenBank. Spike protein sequences from specimens CDAB0017RSV and CDAB0040RSV formed a monophyletic clade, as did those from specimens CDAB0203R and CDAB0217R, reflecting their membership in partial RdRp-based phylogenetic groups W-Beta-2 and W-Beta-3, respectively (*Figure 8*). These novel spike proteins also grouped with spike protein sequences from three *betacoronaviruses* in GenBank: HQ728482.1, MG693168.1, and NC_048212.1 (*Figure 8*). The spike protein sequence from specimen CDAB0492R,

**Figure 9.** Phylogenetic tree of translated spike gene sequences from *betacoronaviruses*. Spike sequences are coloured according to whether they were from study specimens (blue), human coronaviruses (CoVs) (red), RefSeq (black), or GenBank (grey). Only the 25 closest-matching spike sequences from GenBank were included, as determined by blastp bitscores. GenBank and RefSeq accession numbers are provided in parentheses. The scale bar measures amino acid substitutions per site.

**Table 3.** Alignments between translated spike sequences from study specimens and phylogenetically proximate entries from GenBank and RefSeq.

Alignments were conducted with blastp. Reference sequence host and collection location were obtained from GenBank entry summaries.

| Specimen | Specimen host | Reference sequence GenBank accession number | Reference sequence host | Reference sequence collection location | Alignment query coverage (%) | Alignment identity (%) | Alignment positivity (%) |
|---|---|---|---|---|---|---|---|
| CDAB0492R | *Mops condylurus* | HQ728486.1 | *Chaerephon sp.* | Kenya | 100 | 71.2 | 80.1 |
| CDAB0492R | *Mops condylurus* | MZ081383.1 | *Chaerephon plicatus* | Yunnan, China | 100 | 65.8 | 77.5 |
| CDAB0017RSV | *Micropteropus pusillus* | HQ728482.1 | *Eidolon helvum* | Kenya | 99 | 76.5 | 85.7 |
| CDAB0017RSV | *Micropteropus pusillus* | MG693168.1 | *Eidolon helvum* | Cameroon | 99 | 63.7 | 77.7 |
| CDAB0040RSV | *Myonycteris sp.* | HQ728482.1 | *Eidolon helvum* | Kenya | 99 | 75.9 | 84.7 |
| CDAB0040RSV | *Myonycteris sp.* | MG693168.1 | *Eidolon helvum* | Cameroon | 99 | 64.4 | 77.7 |
| CDAB0203R | *Eidolon helvum* | HQ728482.1 | *Eidolon helvum* | Kenya | 100 | 73.7 | 85.3 |
| CDAB0203R | *Eidolon helvum* | MG693168.1 | *Eidolon helvum* | Cameroon | 100 | 65.6 | 78.8 |
| CDAB0217R | *Eidolon helvum* | HQ728482.1 | *Eidolon helvum* | Kenya | 100 | 73.5 | 85.1 |
| CDAB0217R | *Eidolon helvum* | MG693168.1 | *Eidolon helvum* | Cameroon | 100 | 65.2 | 79.0 |

the lone Q-Alpha-4 representative, grouped with spikes from two *alphacoronaviruses* in GenBank: HQ728486.1 and MZ081383.1 (*Figure 9*). None of the CoVs recovered from these specimens were closely related to CoVs that infect humans based on spike gene homology.

Pairwise global alignments of amino acid sequences were conducted between these novel spike genes and the spike genes from GenBank with which they grouped phylogenetically. Alignments completely covered all novel spike sequences, but they were all less than 76.5% identical and less than 85.7% positive (*Table 3*). We compared host species and geographic collection locations for our study specimens and the phylogenetically related spike sequences. Only specimens CDAB0203R and CDAB0217R were collected from the same bat species as their closest spike protein matches in GenBank (*Eidolon helvum*). Other specimens were detected in bat genera different from their closest GenBank match. All study specimens were collected from the DRC, but their closest GenBank

**Table 4.** Nucleotide alignments between novel spike genes from study specimens and phylogenetically related sequences from GenBank and RefSeq.

Alignments were conducted with blastn. Discontinuous alignments are represented as multiple lines in the table, for example, CDAB0217R vs. MG693168.1.

| Specimen | Reference sequence GenBank accession number | Alignment query coverage (%) | Alignment identity (%) |
|---|---|---|---|
| CDAB0492R | HQ728486.1 | 60 | 81.0 |
| CDAB0492R | MZ081383.1 | 18 | 71.5 |
| CDAB0040RSV | HQ728482.1 | 83 | 75.4 |
| CDAB0203R | HQ728482.1 | 78 | 75.5 |
| CDAB0203R | MG693168.1 | 45 | 76.6 |
| CDAB0217R | HQ728482.1 | 71 | 76.0 |
| CDAB0217R | MG693168.1 | 47 | 75.7 |
| CDAB0217R | MG693168.1 | 47 | 84.6 |

matches were collected from diverse locales, including neighbouring Kenya, Cameroon in West Africa, and Yunnan province in China. Taken together, these low alignment scores, disparate host species, and dispersed collection locations suggested these viruses belong to extensive but hitherto poorly characterized taxa of CoV.

We also conducted pairwise global alignments of nucleotide sequences. This was done to confirm that probe capture had been hindered by divergence of these novel spike genes from their closest matches in GenBank, which we had used to design our custom panel. For specimen CDAB0017RSV, sequence similarity was so low that no alignment was generated for the spike gene (*Table 4*). Nucleotide alignments for the other specimens were all incomplete (18–83% coverage of the novel spike sequence) with low nucleotide identities (71.5–84.6%).

## Discussion

This study highlights the potential for probe capture to recover greater extents of CoV genome compared to standard amplicon sequencing methods. In discovery and surveillance applications, this would permit characterization of CoV genomes outside of the constrained partial RdRp regions that are typically described, enabling additional phylogenetic resolution among specimens with similar partial RdRp sequences. Recovering more extensive fragments from diverse regions of the genome would also provide additional genetic sequence to compare against reference sequences in databases like GenBank and RefSeq. This could permit more confident identification of known threats and better assessment of virulence and potential spill-over from novel CoVs. Sequences from additional genome regions could also be used to identify CoVs where recombination has occurred, which is increasing recognized as a potential hallmark of zoonotic CoVs (*Hu et al., 2015*; *Corman et al., 2018*; *Ye et al., 2020*; *Ruiz-Aravena et al., 2021*).

This study also showed the usefulness of probe capture for identifying specimens that warrant the expense of deep metagenomic sequencing for more extensive characterization. The genomic regions missed by the probe panel can provide as much insight into viral novelty as the sequences that are recovered. In this study, failure to capture complete spike gene sequences, even from libraries with otherwise extensive coverage, was successfully used to predict the presence of novel spike genes. Furthermore, contiguity across recovered regions can be used to evaluate abundance and intactness of viral genomic material, identifying specimens where deep metagenomic sequencing is likeliest to succeed. This is valuable when targeting higher taxonomic levels where methods for directly quantifying viral genome copies are hindered by the same genomic variability that constrains amplicon sequencing.

This study also revealed two important limitations for probe capture in CoV discovery and surveillance applications. The first, which appeared to be the most limiting in this study, is the in vitro sensitivity of this method. Probe capture must be performed on already constructed metagenomic sequencing libraries. The library construction process involves numerous sequential biochemical reactions and bead clean-ups, where inefficiencies result in compounding losses of input material. Combined with the low prevalence of viral genomic material in swab specimens, these loses of input material can lead to the presence of incomplete viral genomes in sequencing libraries and stochastic recovery during probe capture. Amplicon sequencing does not suffer the same attrition because enrichment occurs as the first step of the process, allowing library construction to occur on abundant amplicon input material. Further work optimizing metagenomic library construction protocols could be done to improve sensitivity for probe capture. Also, this study relied on archived material in suboptimal condition, so better results could be expected from fresh surveillance specimens.

The second limitation highlighted by this work is the challenge of designing hybridization probes from available reference sequences for poorly characterized taxa. Currently, the extent of human knowledge about bat CoV diversity remains limited, especially across hypervariable genes like spike, and it seems impossible to design a broadly inclusive pan-bat CoV probe panel at this moment. As recently as 2017, it was observed that only 6% of CoV sequences in GenBank were from bats, while the remaining 94% of sequences concentrated on a limited number of known human and livestock pathogens (*Anthony et al., 2017*). The vastness of CoV diversity that remains to be characterized is evident by the continuing high rate of novel CoV discovery by research studies and surveillance programs, this current work included (for example *Tao et al., 2017*; *Wang et al., 2017*; *Markotter et al., 2019*; *Wang et al., 2019*; *Nziza et al., 2020*; *Valitutto et al., 2020*; *Kumakamba et al., 2021*;

*Shapiro et al., 2021*; *Tan et al., 2021*; *Wang et al., 2021*; *Zhou et al., 2021*; *Alkhovsky et al., 2022*; *Ntumvi et al., 2022*).

Fortunately, probe capture is highly adaptable and existing panels can be easily supplemented with additional probes as new CoV taxa are described. For instance, the genomes recovered in this study could be used to design supplemental probes for re-capturing existing specimens as well as for future projects with new specimens. Improved recovery would be especially expected for projects returning to similar geographic regions targeting similar bat populations. Additionally, as CoV evolution becomes better understood and modeled, 'predictive' probe panels could be attempted. These panels would interpolate existing genomes to provide coverage of hypothetical extant taxa that have not yet been characterized. Similarly, they could extrapolate to target likely future variants.

Crucially, these probe design limitations are only a meaningful impediment for CoV discovery, specifically the gold standard recovery of complete genomes; surveillance activities do not require recovery of the entire genome to adequately detect known pathogenic threats. Furthermore, extensive sequencing of zoonotic CoV taxa that have already emerged has provided abundant reference sequences for probe design geared towards genomic detection of these known pathogenic threats. Panels could also be expanded to include other zoonotic viral taxa that circulate in bats like paramyxoviruses and filoviruses, thereby streamlining surveillance programs.

Our results lead us to conclude that probe capture amounts to a trade-off; sensitivity limitations mean that CoV sequence recovery may occur less frequently than with amplicon sequencing, but when it does succeed, CoV sequences may be more extense and more diverse. Likewise, probe panel designs may not be broadly inclusive enough to recover complete genomes in all cases, but the sequencing depth required – and thus the cost per specimen – to attempt recovery will be fractional compared to untargeted methods. Consequently, probe capture is not a replacement for amplicon sequencing or deep metagenomic sequencing, but a complementary method to both.

Based on these observations, we propose that the most effective CoV discovery and surveillance programs will combine amplicon sequencing, probe capture, and deep metagenomic sequencing. The simplicity, sensitivity, and affordability of amplicon sequencing make it well suited for initial screening. This method also requires the least laboratory infrastructure, much of which already exists in surveillance hotspots at facilities with extensive experience and established track records of success. Screening by amplicon sequencing would enable direct phylogenetic comparisons between specimens across consistent genomic loci and enable a preliminary assessment of threat and novelty. This screening would also identify CoV-positive specimens warranting further study, limiting the number of specimens to be transported to more specialized laboratories with probe capture and deep sequencing capacity.

Probe capture on select CoV-positive specimens would be valuable for potentially acquiring additional sequence information which could refine assessments of threat and novelty. As new CoVs are characterized and probe panel designs are expanded, recovery of host range and virulence factors by probe capture would steadily increase.

Finally, probe capture results would be used to identify interesting specimens warranting the expense of deep metagenomic sequencing. It would also be used to triage specimens based on the abundance and intactness of viral genomic material inferred from the probe capture results. Deep sequencing would allow for the most extensive characterization and evaluation of novel CoV genomes, especially for hypervariable host range and virulence factors like spike gene. It would also provide novel sequences for updating probe panel designs. Deploying these methods in conjunction, with each used to its strength, would enable highly effective genomics-based discovery and surveillance for bat CoVs.

## Materials and methods
### Bat swab specimens and partial RdRP sequences

As part of a previous study, rectal and oral swabs were collected from bats in DRC between August 2015 and June 2018 (*Kumakamba et al., 2021*). The previous study conducted CoV screening of these swabs using two consensus PCR assays targeting small regions in the RNA-dependant RNA polymerase (RdRP) gene of bat *alpha*- and *betacoronaviruses* (*Quan et al., 2010*; *Watanabe et al., 2010*). The previous study also Sanger sequenced these amplicons for CoV phylogenetic characterization.

For the current study, aliquots of remaining material from 21 of these swab specimens were shipped to Canada: RNA extracts and swab transport medium were provided for 4 specimens, swab transport medium only was provided for 2 specimens, and RNA extracts only were provided for 15 specimens. Swab transport medium aliquots were re-extracted upon arrival in Canada using the Invitrogen TRIzol Reagent (#15596026) following the manufacturer's protocol. RNA concentration and RIN for all RNA extracts were measured using the Agilent BioAnalyzer 2100 instrument with the RNA 6000 Nano kit.

### Probe panel design and reference sequence coverage assessments

All available bat CoV sequences were downloaded from NCBI GenBank on 4 October 2020. A custom panel of 20,000 hybridization probes was designed from these sequences using the ProbeTools package (v0.0.5) (*Kuchinski et al., 2022c*). All available sequences in the following taxa were downloaded from NCBI GenBank on 4 October 2020: unclassified *coronavirinae* (txid: 693995), unclassified *coronaviridae* (txid: 1986197), *alphacoronavirus* (txid: 693996), and *betacoronavirus* (txid: 694002). Bat CoV sequences were extracted by searching sequence headers for bat-related key words identified by the authors. These sequences were used as targets for probe design with the ProbeTools package (v0.0.5) (https://github.com/KevinKuchinski/ProbeTools; copy archived at swh:1:rev:20f-78c3af2e88be28ac6130b3588f5c16e49c7a6; *Kuchinski et al., 2022c*; *Kuchinski, 2022b*). All possible probes were generated from the bat CoV sequences using the *makeprobes* module with a batch size of 100 probes. This generated a core panel of 18,365 probes.

Since the next breakpoint in the manufacturer's pricing occurred at 20,000 probes, we designed additional probes targeting conserved motifs in CoVs from non-bat hosts. We used the *capture* and *getlowcov* modules to extract regions of the unclassified *coronavirinae*, unclassified *coronaviridae*, *alphacoronavirus*, and *betacoronavirus* sequences from all hosts not already covered by the core panel. These regions were then used as input targets for *makeprobes* with a batch size of 50 probes. The first 1605 probes generated in this way became the supplemental panel. While designing the supplemental panel, we removed SARS-CoV-2 sequences from the *betacoronavirus* space because they were over-represented and could have biased probe design towards this single taxon. To ensure coverage of SARS-CoV-2-related viruses by our panel, we used the *capture* and *getlowcov* modules to extract regions of the Wuhan-Hu-1 reference genome (MN908947.3) not already covered by the core panel. These regions were then used as input targets for *makeprobes* with a batch size of 1 probe, generating 29 probes that were added to the supplemental panel.

The following were combined to create the final panel: the core panel of 18,365 probes generated from bat CoV sequences, the supplemental panel of 1634 probes targeting conserved motifs in non-bat CoVs and SARS-CoV-2, and a single probe targeting our artificial control oligo sequence. The final panel (*Supplementary file 1*) was synthesized by Twist Bioscience (San Francisco, CA, USA). Probe coverage of reference sequences was assessed in silico using ProbeTools.

### Library construction and pooling

Sequencing libraries were constructed using the NEBNext Ultra II RNA Library Prep with Sample Purification Beads kit (E7775). Five µl of undiluted RNA specimen was used as input for first strand synthesis. The fragmentation reaction incubation was shortened to 2 min at 94°C while the first strand synthesis incubations were modified to 10 min at 25°C, followed by 50 min at 42°C, followed by 10 min at 70°C. Second strand synthesis, bead clean-up, and end prep reactions were performed according to the kit's protocol. The adapter ligation incubation was extended to 60 min at 20°C, and the USER digest was also extended to 60 min at 37°C. Following another bead clean-up performed according to the kit protocol, libraries were barcoded with NEBNext Multiplex Oligos for Illumina (96 Unique Dual Index Primer Pairs) kit (E6440). Barcoding PCRs used the following cycling conditions: 1 cycle of 98°C for 1 min; 12 cycles of 98°C for 30 s, then 65°C for 75 s; 1 cycle of 65°C for 10 min. Barcoded libraries were purified with the final bead clean-up according to the kit's protocol.

### Probe capture

Libraries were quantified with the Invitrogen Qubit dsDNA HS kit (Q32851), then 180 ng of each library was pooled together. The library pool was fully evaporated in a GeneVac miVac DNA concentrator (DNA-12060-C00) instrument. The dried library pool used to set up a hybridization reaction with 0.2 fmol/probe of our custom bat CoV probe panel (Twist Biosciences, San Francisco, CA,

USA), Twist Universal Blockers (#100578), and the Twist Fast Hybridization Reagents kit (#101174) following the manufacturer's protocol. The pool was captured twice sequentially by our custom probe panel. Hybridization reactions were incubated at 70°C for 16 hr, then captured and washed with the Twist Binding and Purification Beads (#100983) and Twist Fast Hybridization Wash buffers (#101025) following the manufacturer's protocol until the final step, at which point the streptavidin bead slurry was resuspended in 22.5 µl of nuclease-free water instead of 50 µl. The entire 22.5 µl volume was used in the post-capture PCR, which was set up with NEBNext Ultra II Q5 2X Master Mix (#M0544), and Illumina amplification primers from the Twist Fast Hybridization Reagents kit (#101174). Post-capture PCRs were conducted with the following cycling conditions: 1 cycle of 98°C for 60 s; 25 cycles of 98°C for 30 s, then 60°C for 30 s, then 65°C for 75 s; 1 cycle of 65°C for 10 min. Post-capture PCRs were purified using ×0.8 SPRI beads from the Twist Binding and Purification Beads (#100983). Bead clean-up reactions were washed twice with 200 µl of 80% ethanol and eluted in 20 µl of nuclease-free water. Following the first capture, the captured pool was again completely evaporated, then a second capture was performed as before.

Control specimens were prepared by spiking 100,000 copies of a synthetic control oligo into 200 ng of Invitrogen Human Reference RNA (#QS0639). The control oligo was manufactured by Integrated DNA Technologies (Coralville, IA, USA) as a dsDNA gBlock with a known artificial sequence created by the authors. Probes targeting the control oligo were included in the custom capture panel. Control specimens were prepared into libraries alongside bat specimens from the same reagent master mixes, and they were included in the same pool for probe capture.

## Sequencing of captured libraries and removal of index hop artefacts

Probe captured libraries were sequenced on an Illumina MiSeq instrument using V2 300 cycle reagent kits (#MS-102-2002). The double-captured library pool was sequenced across two MiSeq runs. The first run generated paired-end reads where each end was sequenced with 150 cycles. The second run generated paired-end reads where the first end was sequenced with 15 cycles and the second end was sequenced with 285 cycles. Index hops were filtered from both runs using HopDropper (v0.0.3) (https://github.com/KevinKuchinski/HopDropper; copy archived at swh:1:rev:12b9e4e5510f-d1c202d3e74a291a12d62eeafe37; *Kuchinski, 2022a*) with UMIs of length 14, requiring a minimum base quality of PHRED 30, and discarding UMI pairs appearing only once. After removing index hops, reads from the second MiSeq run were treated as single-ended. This was done by discarding the short first end which was only necessary for index hop removal by HopDropper.

Detection and enrichment of the control oligo sequence in control specimen libraries was used as a positive control for library construction and probe capture. Absence of control oligo sequences in bat specimen libraries and absence of bat CoV sequences in control specimen libraries were used as a negative control for contamination and as a positive control for index hop removal by HopDropper (v0.0.3) (https://github.com/KevinKuchinski/HopDropper; *Kuchinski, 2022a*).

## De novo assembly of contigs from captured reads

coronaSPAdes (v3.15.0) was used to assemble contigs de novo from probe captured MiSeq data (*Meleshko et al., 2021*). Reads from the first MiSeq run were provided to coronaSPAdes as paired-end data, while reads from the second MiSeq run were provided as single-end data. CoV contigs were identified using BLASTn (v2.12.0) against a local database composed of all *coronaviridae* sequences (txid: 11118) in GenBank available as of 11 October 2021 (*Camacho et al., 2009*).

## Alignment of reads and contigs to bat CoV reference sequences

Probe captured reads were mapped to selected reference genomes using bwa mem (0.7.17-r1188). Alignments were filtered with samtools view (v1.11) to retain properly paired reads (bitflag 3) and exclude unmapped reads, reads without mapped mates, not primary alignments, supplementary alignments, and reads failing platform/vendor quality checks (bitflag 2828) (*Li and Durbin, 2009a*, *Li et al., 2009b*). Samtools sort and index (v1.11) were then used to sort and index filtered alignments. Depth and extent of read coverage were determined with bedtools genomecov (v2.30.0) (*Quinlan and Hall, 2010*). Contig coverage was determined by aligning contigs to reference sequences with BLASTn (v2.12.0) and extracting subject start and subject end coordinates (*Camacho et al., 2009*).

**Table 5.** RT-qPCR primer sequences.

| Specimen | Primers | Primer sequences | Standard curve $R^2$ |
|---|---|---|---|
| CDAB0146R-PRE<br>CDAB0158R-PRE<br>CDAB0160R-PRE<br>CDAB0173R-PRE<br>CDAB0174R-PRE<br>CDAB0203R-PRE<br>CDAB0212R-PRE<br>CDAB0217R-PRE | Beta-3_rdrp_FWD<br>Beta-3_rdrp_REV | ATA TAT GTC AGG CCG TTA GTG C<br>CCA TAT AGA GGC GAT GTT GC | 0.995 |
| CDAB0486R-PRE<br>CDAB0488R-PRE<br>CDAB0488R-TRI<br>CDAB0491R-PRE<br>CDAB0491R-TRI<br>CDAB0492R-PRE<br>CDAB0492R-TRI<br>CDAB0494O-TRI-PRE<br>CDAB0494R-PRE<br>CDAB0494R-TRI<br>CDAB0495O-PRE<br>CDAB0495R-TRI | Alpha_4_rdrp_FWD<br>Alpha_4_rdrp_REV | GCG ACT ACC TGG TAA ACC TAT C<br>CTT TGC CGC ACT CAC AAA C | 0.989 |
| CDAB0017R-PRE<br>CDAB0040R-PRE<br>CDAB0040RSV-PRE | Beta-2_rdrp_FWD<br>Beta-2_rdrp_REV | CAC TAC TTG TAC CAC CAG GTT T<br>TTG TAG TGG TTC TGA TCG TTT T | 0.998 |
| CDAB0305R-PRE | D0305_rdrp_FWD<br>D0305_rdrp_REV | GAC GGC AAT AAG GTG CAT AAC<br>AGT CAG AAA CCA AGT CCT CAT C | 0.999 |
| CDAB0113RSV-PRE | D0113_rdrp_FWD<br>D0113_rdrp_REV | GTA CGT TGA GTG AGC GGT ATT<br>GAT GAA GTT CCA CCT GGC TTA | 0.998 |

## RT-qPCR measurement of CoV abundance in specimens

Quantitative PCRs were conducted in duplicate for each RNA sample using the Luna Universal One-step RT-qPCR kit (New England Biolabs Inc, MA, USA) and 400 nM of the forward and reverse primers for 40 cycles. Custom primers (*Table 5*) were designed based on previously generated partial RdRP sequences from these specimens (*Kumakamba et al., 2021*). To test for primer dimers, melt curves were performed for all primer sets and no template control reactions were also conducted. Genome abundance was estimated by creating standard curves of synthetic gBlocks (*Supplementary file 2*) containing the partial RdRP sequences (Integrated DNA Technologies Ltd., IA, USA). Standard curves were produced for each primer using six serial 10-fold dilutions of gBlocks (Integrated DNA Technologies Ltd., IA, USA) as template. Copies per microliter were calculated by multiplying the concentration (ng/µl) of the resuspended gBlocks by the molecular weight (fmol/ng), by $1\times10^{-15}$ mol/fmol, and by Avogadro's number ($6.022\times10^{23}$) as recommended by the manufacturers (https://www.idtdna.com/pages/education/decoded/article/tips-for-working-with-gblocks-gene-fragments). Standards were run alongside samples using the Luna Universal qPCR Master Mix kit (New England Biolabs Inc, MA, USA) and 250 nM of each primer. qPCR was performed using a StepOnePlus Real-Time PCR System (Applied Biosystems, CA, USA) using the recommended thermocycler settings from the Luna Universal One-step RT-qPCR kit. Quantities (copies per microliter) for each sample were calculated in the StepOnePlus software v2.3 according to the standard curves included in each qPCR run while accounting for dilutions.

## Deep metagenomic sequencing of uncaptured libraries and generation of complete viral genomes

New libraries were prepared from selected specimens following the same protocol as for libraries that were probe captured. These libraries were sequenced on an Illumina HiSeq X instrument by the Michael Smith Genome Sciences Centre (Vancouver, BC, Canada). Reads were assembled and scaffolded into draft genomes with coronaSPAdes (v3.15.3) (*Meleshko et al., 2021*). CoV-sized scaffolds

were manually inspected to identify draft genomes. For one specimen (CDAB0492R), two contigs were manually joined to complete a complete draft genome.

HiSeq reads were mapped to draft genomes using bwa mem (v0.7.17-r1188). Alignments were filtered with samtools view (v1.11) to retain properly paired reads (bitflag 3) and exclude unmapped reads, reads without mapped mates, not primary alignments, supplementary alignments, and reads failing platform/vendor quality checks (bitflag 2828) (*Li and Durbin, 2009a*, *Li et al., 2009b*). Samtools sort and index (v1.11) were then used to sort and index filtered alignments. Variants were called with bcftools mpileup and call (v1.9) (*Danecek et al., 2021*). For bcftools mpileup, 30 was used as the minimum read mapping (-q) and base quality scores (-Q), and a minimum of 10 gapped reads was used for indel candidates (-M). For bcftools call, a ploidy of 1 was used (--ploidy). Low coverage positions in the draft genomes (<10 reads) were masked using bedtools genomecov (v2.30.0) (*Quinlan and Hall, 2010*), then variants were applied to draft genomes with bcftools consensus (v1.9) to generate final complete genomes (*Danecek et al., 2021*).

## Phylogenetic analysis of novel spike gene sequences

Novel spike gene coding sequences were identified in three steps. First, we obtained the regions annotated as spike gene coding sequences from each study specimen's closest reference sequence in GenBank/RefSeq. Second, these spike coding sequences from the closest reference sequences were aligned to the final genomes of the novel bat CoVs using BLASTn (v2.12.0) (*Camacho et al., 2009*). Third, novel spike coding sequences were extracted using the subject start and end coordinates from the alignment. Novel spike CDSs were then translated using a custom Python script. Translated sequences were queried against all translated *coronaviridae* spike sequences in GenBank (available on 11 October 2021) using BLASTp (v2.12.0) (*Camacho et al., 2009*). For each genus, novel spike genes from study specimens were combined with the 25 closest-matching GenBank spike sequences (based on alignment bitscore) and all spike sequences available in RefSeq. Multiple sequence alignments were conducted with clustalw (v2.1) with default parameters, then phylogenetic trees were constructed from aligned sequences using PhyML (v3.3.20190909) with 100 bootstrap replicates (*Thompson et al., 1994*; *Guindon et al., 2005*).

## Acknowledgements

The authors would like to thank: members of the Institute for Microbial Systems and Society, Caroline Cameron, and David Alexander for helpful discussions; the government of the DRC for the permission to conduct this study and the late Prime Mulembakani for his invaluable contribution to the success of this work; Guy Midingi Sepolo, Joseph Fair, Bradley Schneider, Anne Rimoin, Nicole Hoff, and other members of the PREDICT consortium for their support. This study was made possible by funding from Genome Prairie COVID-19 Rapid Regional Response (COV3R) and the Saskatchewan Health Research Foundation COV3R Partnership grants. This study was made possible partially thanks to the generous support of the American people through the USAID Emerging Pandemic Threats PREDICT program (cooperative agreement number AID-OAA-A-14-00102). The contents are the responsibility of the authors and do not necessarily reflect the views of USAID or the United States Government. Funding sources were not involved in study design, data collection and interpretation, or the decision to submit the work for publication. We applied the Contributor Roles Taxonomy (CRediT) plus standard attribution practices in biological sciences for ordering the author list. Authors in Canada: conceived the study, developed the laboratory methods used, conducted all laboratory work, analysed all molecular data, and wrote the manuscript. Authors in the Democratic Republic of Congo and Cameroon: provided specimens that had been collected for a previously published study (*Kumakamba et al., 2021*), consented to sharing these specimens, and reviewed the manuscript. Authors in the United States of America designed and acquired funding for the PREDICT program, oversaw specimen and data transfer through PREDICT, and reviewed the manuscript.

## Additional information

### Competing interests

Charles Kumakamba, Francisca Muyembe, Placide Mbala Kingebeni, Ipos Ngay Lukusa, Frida N'Kawa, Joseph Atibu Losoma, James A Ayukekbong, Corina Monagin, Edward M Rubin, David J McIver: were employed by Metabiota Inc. Maria Makuwa, Karen Saylors, Christian E Lange: are employees of Labyrinth Global Health Inc and were employed by Metabiota Inc. Amethyst Gillis: is an employee of Development Alternatives Inc and was employed by Metabiota Inc. Damien O Joly: is an employee of Nyati Health Consulting and was employed by Metabiota Inc. Nathan D Wolfe: is an employee of Metabiota Inc. The other authors declare that no competing interests exist.

### Funding

| Funder | Grant reference number | Author |
|---|---|---|
| Genome Canada | COV3R | Natalie A Prystajecky |
| Saskatchewan Health Research Foundation | COV3R | Andrew DS Cameron |
| United States Agency for International Development | AID-OAA-A-14-00102 | Karen Saylors |

The funders had no role in study design, data collection and interpretation, or the decision to submit the work for publication.

### Author contributions

Kevin S Kuchinski, Conceptualization, Software, Formal analysis, Investigation, Visualization, Methodology, Writing - original draft, Writing – review and editing; Kara D Loos, Danae M Suchan, Jennifer N Russell, Ashton N Sies, Formal analysis, Investigation, Writing – review and editing; Charles Kumakamba, Supervision, Investigation; Francisca Muyembe, Ipos Ngay Lukusa, Frida N'Kawa, Joseph Atibu Losoma, Nicole A Lerminiaux, Investigation; Placide Mbala Kingebeni, Data curation, Supervision; Maria Makuwa, Supervision, Investigation, Project administration; Amethyst Gillis, Data curation, Formal analysis, Validation; Matthew LeBreton, Data curation, Investigation; James A Ayukekbong, Data curation, Project administration; Corina Monagin, Project administration; Damien O Joly, Karen Saylors, Supervision, Funding acquisition, Project administration; Nathan D Wolfe, Funding acquisition; Edward M Rubin, Supervision, Project administration; Jean J Muyembe Tamfum, Resources, Funding acquisition; Natalie A Prystajecky, Supervision; David J McIver, Data curation, Formal analysis, Project administration, Writing – review and editing; Christian E Lange, Data curation, Formal analysis, Supervision, Validation, Writing – review and editing; Andrew DS Cameron, Conceptualization, Formal analysis, Supervision, Funding acquisition, Project administration, Writing – review and editing

### Author ORCIDs

Kevin S Kuchinski (ID) http://orcid.org/0000-0001-7588-4910
Ashton N Sies (ID) http://orcid.org/0000-0003-2145-7010
David J McIver (ID) http://orcid.org/0000-0002-9507-1674
Christian E Lange (ID) http://orcid.org/0000-0002-0664-9367
Andrew DS Cameron (ID) http://orcid.org/0000-0003-1560-8572

### Decision letter and Author response

Decision letter https://doi.org/10.7554/eLife.79777.sa1
Author response https://doi.org/10.7554/eLife.79777.sa2

## Additional files

### Supplementary files

• Supplementary file 1. Probe sequences, fasta format.

• Supplementary file 2. gBlock sequences, fasta format. Two gBlocks were generated for this study. 'Gblck_Beta2_17_40_B3_Alpha' is a composite of RdRp nucleotide sequences from W-Beta-2 (samples CDAB0017 and CDAB0040), W-Beta-3, and Q-Alpha-4 (*Table 1*). 'Gblck_Beta2_0305_B4_Alpha' is a composite of RdRp nucleotide sequences from W-Beta-2 (samples CDAB0305), W-Beta-4, and Q-Alpha-4. The Q-Alpha-4 sequence is identical in both gBlocks.

• MDAR checklist

## Data availability

The sequence data from this study is available at National Center for Biotechnology Information (NCBI) Sequence Read Archive (SRA) as BioProject PRJNA823716 (https://www.ncbi.nlm.nih.gov/bioproject/PRJNA823716/). The assembled coronavirus genomes are available at GenBank with following accession numbers: ON313743 (CDAB0017RSV); ON313744 (CDAB0040RSV); ON313745 (CDAB0203R); ON313746 (CDAB0217R); ON313747 (CDAB0492R).

The following dataset was generated:

| Author(s) | Year | Dataset title | Dataset URL | Database and Identifier |
|---|---|---|---|---|
| Kuchinski KS, Loos KD, Suchan DM, Russell JN, Sies AN, Cameron ADS | 2022 | Targeted genomic sequencing with probe capture for discovery and surveillance of coronaviruses in bats | https://www.ncbi.nlm.nih.gov/bioproject/?term=PRJNA823716 | NCBI BioProject, PRJNA823716 |

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
