## [Editor Report]

This work applies hybrid-capture sequencing for coronavirus (CoV) surveillance in bats. Given that bats are a major reservoir for animal-to-human virus spillover events, which have caused several major epidemics/pandemics, this is a very important field of research. The reported hybrid-capture method shows some clear advantages over amplicon-based viral sequencing, which is the established standard in the field. This new approach has clear merits that are well supported by the data presented and is likely to become an important tool in viral surveillance programs that ultimately aim to predict/prevent/prepare for future pandemics. The work will be of interest to microbiologists, particularly those studying viruses or interested in genomics surveillance.

---

## [Decision Letter]

**Decision letter after peer review:**

Thank you for submitting your article "Targeted genomic sequencing with probe capture for discovery and surveillance of coronaviruses in bats" for consideration by *eLife*. Your article has been reviewed by 2 peer reviewers, and the evaluation has been overseen by a Reviewing Editor and Bavesh Kana as the Senior Editor. The following individual involved in the review of your submission has agreed to reveal their identity: Ira Deveson (Reviewer #1).

Essential revisions:

1. Please include a simple table of sequencing summary statistics for the study – eg number of sequencing reads overall, on/off target rates, PCR duplicate rates, etc in each library. This is useful for the reader, and some of these basic performance metrics might also help explain the failure to detect expected CoV sequences in some samples.

2. What was the sequence similarity between RdRp contigs obtained via amplicon vs capture sequencing on matched samples? Did capture sequencing always recover the same/similar RdRp sequence, or were there any discordant results? Some text on this would help.

3. The examples of coverage plots in Figure 3 are useful and help to orientate the reader. However, some global summaries of coverage breadth and depth should also be shown, rather than just individual examples. A simple dot plot or bar chart showing the % genome coverage for each of the specimens would be good. And a similar figure showing % coverage for each relevant gene (eg RdRp, spike, etc) would also be informative. The spike generally has poor coverage in the read-depth tracks, it would be good to show this in a clear, simple plot covering all specimens.

4. The hybrid-capture panel design was obviously limited due to the diversity, quality, and completeness of available bat CoV genomes. There are a few open-ended questions on this that could be worth discussing:

Would there be any merit in including known CoV genomes from outside bats on the panel? This might help pick up inter-species transmission events. Please discuss.

Did the authors consider elevating the probe density, or modifying probe sizes, within hypervariable regions, specifically the spike protein? There may well be some technical optimisations that could deliver better performance.

Is it possible to design probes in this region that target imagined (predicted/modelled), but not previously observed, spike-protein sequences? For example, include additional probes that cover a diversity of semi-random spike mutations that could foreseeably occur – an agnostic diversity capture approach.

There is also no reason why other viruses can't be included on the panel. Bats must also carry other informative viruses like influenzas that might be worth looking for. Please discuss.

5. A quantification of the viral load of the samples would be useful to help understand the similarities and differences between the samples post sequencing. For example, SARS-CoV-2 hybrid capture needs a Ct of < 25 to get a reasonable genome whereas ARTIC amplicon sequencing can get similar results at 30. The complexities of working with archival samples and transporting them over long distances limit what can be done.

6. From the methods description of the controls and bioinformatics analysis of the sequencing data, there appears to have been cross-contamination during sample preparation for sequencing. There was then an informatics salvage operation on the data. These salvage operations are fraught with danger, particularly where you have very low levels of RNA and are performing assemblies with very low depths of coverage, which is the case in this work. As the original data, as it came off the sequencer, is not available to the public it is not possible for anyone outside of the study to quantify. This would be a QC fail in SARS-CoV-2 sequencing labs and with a repeat from scratch.

Were NTCs/blanks used for sequencing at the same time as the rest of the samples?

Were there any CoV reads in these NTCs control (before any read scrubbing)?

Many labs have undertaken SARS-CoV-2 work (sequencing, diagnostics, reagent manufacturers) and there is a widespread low level of background contamination. As an indication of background contamination, how many SARS-CoV-2 reads were present in the read data (before any read scrubbing)?

The risk is that the cross-contamination from one overperforming sample can overwhelm an underperforming sample, giving you an erroneous mixed assembly.

Other comments:

1. A positive benefit of amplicon sequencing that should be highlighted is the ability to detect intrahost viral populations.

2. Please check the version of blast used because the version in the text is quite old (possibly just a copy and paste typo).

3. The assembly sizes are small and marginally larger than amplicon sequencing and one can sporadically get different regions of the genome making comparative analysis challenging. This method should really shine on fresh, high viral load samples, so it would be interesting to see it in action, perhaps in the field with sequencing on a MinION. Some comments on this would be useful.

4. The introduction could do a better job of linking back to the literature on the use of hybrid capture for virus sequencing. One paper that comes to mind is https://f1000research.com/articles/4-1062 with the method used for SARS-CoV-2 to great success. There are a lot of papers using hybrid capture for SARS-CoV-2 over the past 2 years demonstrating the relevance of the approach. Please cite the appropriate literature.

Concerns related to how credit has been apportioned to authors, particularly those from DRC.

We acknowledge your earlier correspondence, detailing the roles of authors. It is also acknowledged that some aspects of primary research related to these samples have been published in prior work, with appropriate credit provided to the DRC team. That said, this manuscript makes mention of some primary specimens being shipped to Canada (Specific text: "21 unique specimens were shipped to Canada: 15 as RNA extracts only, 2 as unextracted swabs in transport medium, and 4 as both previously extracted RNA and unextracted swabs in transport medium"). The unextracted swabs would classify as primary specimens. I'm sure you appreciate that collection of such material is a complex process and takes much effort, the current study would not have been possible without these primary specimens. Without knowledge of the intimate workings of your research group, the authorship line up where DRC authors who undertook this collection but do not share primary/senior authorship, emerged as a concern.

We accept your explanation and note your suggestion to include a more detailed author statement. Please do so. That said, there may be merit in discussing this with your team to determine if the author list best reflects the overall intent of the research, including the partnerships created and what appears to be an excellent collaborative relationship between diverse groups, spanning continents. Considerations of joint (sometimes with more than two authors) primary authorship or senior authorship may best reflect the vibrant and collegial nature of these relationships. As a journal, we cannot be prescriptive, the choice is ultimately up to you and your team but I hope this narrative has provided some guidance.

*Reviewer #2 (Recommendations for the authors):*

The authors need to take a long hard look at how this research was conducted and how it got so far without being addressed. Why are the first 5 authors and last 4 authors all from wealthy countries (primarily Canada), with DRC authors dumped in the middle? It is ethically and morally wrong to undertake this kind of colonial research. You should be building skills and capacity in DRC, not just taking samples thousands of miles away and giving tokenistic authorship to local scientists. These kinds of abusive research practices brought about the Nagoya Protocol.

---

## [Author Response]

Essential revisions:1. Please include a simple table of sequencing summary statistics for the study – eg number of sequencing reads overall, on/off target rates, PCR duplicate rates, etc in each library. This is useful for the reader, and some of these basic performance metrics might also help explain the failure to detect expected CoV sequences in some samples.

We have prepared a table summarizing the number of reads and total bases sequenced for each library. We have provided these statistics for the raw output, the valid data, *i.e.* after pre-processing to trim adapters and low quality bases and to remove index hops and chimeras. For each library, we have also estimated the number of on-target reads and their total bases by mapping valid reads to the CoV contigs generated from that library and the reference sequence selected for that library. These data have been provided as Table 2, which is now referenced in the text at Line 140 and 176 to 178.

2. What was the sequence similarity between RdRp contigs obtained via amplicon vs capture sequencing on matched samples? Did capture sequencing always recover the same/similar RdRp sequence, or were there any discordant results? Some text on this would help.

For each specimen, we aligned the previously published partial RdRP sequence (generated by amplicon sequencing using the Watanabe or Quan assays) against the contigs generated from the probe captured libraries using de novo assembly. For specimens where the partial RdRP sequence was assembled, nucleotide identities ranged from 99.3% to 100% (median=100%, maximum 2 mismatches). We have included this information in the text at Lines 154 to 156.

3. The examples of coverage plots in Figure 3 are useful and help to orientate the reader. However, some global summaries of coverage breadth and depth should also be shown, rather than just individual examples. A simple dot plot or bar chart showing the % genome coverage for each of the specimens would be good. And a similar figure showing % coverage for each relevant gene (eg RdRp, spike, etc) would also be informative. The spike generally has poor coverage in the read-depth tracks, it would be good to show this in a clear, simple plot covering all specimens.

We thank the reviewers for this suggestion and have incorporated a dot plot showing percent coverage of the spike and RdRp genes across all specimens (Figure 4C). We have added text to describe these data at Lines 192 to 197. We have also updated the legend for Figure 4.

The reviewers have also suggested a similar plot showing the percent coverage of the whole genome. We considered making this plot while preparing the manuscript, but we opted to present the extent of recovery in absolute terms (the number of nucleotides) instead (Figure 4A). This decision was made for a few reasons. First, quite simply, we could not calculate the percent coverage because we did not know the true size of the genomes; we had no denominator for these calculations. We considered estimating the total size using the reference sequences selected for each phylogenetic group. This was problematic, however, because the best reference sequences (*i.e.* the ones that allowed us to identify the most CoV sequence) were not always complete genomes.

We also decided that absolute numbers of nucleotides were more meaningful for this exercise. Since we are dealing with multiple CoV taxa and CoV genomes vary in size, percentages of coverage are not necessarily comparable recovery metrics between phylogenetic groups. For instance, 10% recovery of two different CoV taxa may represent significantly different amounts of genome recovered. Furthermore, for bioinformatic identification of unknown sequences, the number of nucleotides queried is more relevant than what percentage of their genome they might represent. Thus, we reasoned that presenting Figure 4A with absolute recoveries was more useful for assessing the bioinformatic utility of these results in a discovery/surveillance application.

These reasons (unknown genome lengths for novel taxa, genome length variation between taxa, and the importance of absolute query size for bioinformatic identification) are why amplicons used for phylogenetic studies (*e.g.* Watanabe and Quan) are described using their amplicon size and not the percentage of some genome that those amplicons represent. This raises the final reason we expressed recovery in terms of total nucleotides recovered: since we were comparing our results to the Watanabe and Quan amplicons, we wanted to express our data in the same units.

4. The hybrid-capture panel design was obviously limited due to the diversity, quality, and completeness of available bat CoV genomes. There are a few open-ended questions on this that could be worth discussing:Would there be any merit in including known CoV genomes from outside bats on the panel? This might help pick up inter-species transmission events. Please discuss.

The probe design strategy proposed by the reviewers certainly has merit, and it is one that we implemented when designing this panel. As described in the methods, our initial bat-focused design generated 18,365 probes (Lines 434 to 437; text was formerly in a supplement, now in main text). The next breakpoint in the manufacturer’s pricing was at 20,000 probes, so we decided to fill out the panel with probes designed against other *α-* and *betacoronaviruses* from non-bat hosts. If our budget for probes had been larger, we would have extended this concept further and made a broader pan-*α-* and *betacoronavirus* panel.

Did the authors consider elevating the probe density, or modifying probe sizes, within hypervariable regions, specifically the spike protein? There may well be some technical optimisations that could deliver better performance.

Probe design was conducted using the ProbeTools software (https://github.com/KevinKuchinski/ProbeTools). This software applies an incremental k-mer clustering algorithm to optimize coverage of hypervariable targets while minimizing the number of redundant probes. This software essentially performs the same kind of probe selection optimizations a human would attempt manually, except much faster and more systematically. These algorithms and their implementation are described in the ProbeTools manuscript (Kuchinski *et al.,* 2022). The ProbeTools manuscript also provides *in silico* and in vitro validation of the software’s probe design optimization algorithms. When we submitted the current bat CoV manuscript, we cited a preprint of the ProbeTools manuscript. It has since been published (with some valuable revisions that are germane to the reviewers’ comments), so we have updated our bibliography with the citation for the published version. Furthermore, Table 4 suggests that additional attempts to optimize probe design would not have provided much benefit: increased probe density of existing, known spike sequences would not have impacted coverage of these divergent spike genes. We believe the most effective way to broaden the inclusivity of these panels would be to incorporate newly-characterized CoV taxa into the design space, as we mentioned in our discussion (Lines 364 to 369).

Is it possible to design probes in this region that target imagined (predicted/modelled), but not previously observed, spike-protein sequences? For example, include additional probes that cover a diversity of semi-random spike mutations that could foreseeably occur – an agnostic diversity capture approach.

The reviewers raise a very intriguing line of thought. This is a strategy one of the manuscript’s authors previously explored when designing probes for a different project. The idea was to create a target space of amino acid sequences (including alternate sequences containing biochemically similar residues at identified mutable positions). All possible nucleotide spellings of these amino acid sequences (and their alternates) would be enumerated then used as the design space for probes. The rationale was that conservation of protein function would constrain future evolution and the most likely genetic variants would accumulate mostly silent or, at least, homologous mutations.

This strategy quickly became mathematically impractical. For example, the SARS-CoV-2 spike protein sequence contains ~1273 amino acid residues. Since there are roughly 3 alternate trinucleotide codons for each amino acid, that means there are 3^1273^ different nucleotide sequences that encode the same SARS-CoV-2 spike protein. That would inflate the probe design space by a factor of roughly 2.4 x 10^607^! And that is just for one SARS-CoV-2 spike protein sequence… now imagine including all the SARS-CoV-2 spike protein variants, not to mention the numerous, diverse spike proteins of other coronaviruses. The vastness of the space that evolution can explore is truly humbling.

For the sake of argument, let’s say we were going to be more modest in our ambitions: we would degenerate only 5 amino acids positions per spike protein sequence, and we would do this only for the 207 full-length bat CoV sequences in the design space we used for this study. That would still expand the design space by a factor of approximately 50,301-fold (3^5^ x 207). Aside from a limited probe budget, this much larger design space would have likely strained our available computing power, even on high performance servers.

And then there’s the question of whether this would be necessary to accommodate mutations affecting isolated amino acid residues. The probes are 120 nucleotides long. Isolated, sporadic SNPs causing amino acid substitutions might not interfere with hybridization capture if they are flanked by conserved sequence the probes can anneal.

All that being said, we are still excited by this line of inquiry. In the future, machine learning might allow hyper-accurate identification of relevant mutable positions and their most plausible amino acid substitutions. Together with advances in quantum computing, these vast evolutionary spaces might be shrunk and more quickly analyzed. Since we appreciate these invitations to speculate about the future of probe capture technology, we have added the following to our Discussion section (Lines 369 to 372):

“Additionally, as CoV evolution becomes better understood and modeled, “predictive” probe panels could be attempted. These panels would interpolate existing genomes to provide coverage of hypothetical extant taxa that have not yet been characterized. Similarly, these panels could extrapolate to target future variant taxa.”

There is also no reason why other viruses can't be included on the panel. Bats must also carry other informative viruses like influenzas that might be worth looking for. Please discuss.

The reviewers are correct, and the ProbeTools software could be used to facilitate these kinds of designs (some authors of this manuscript are currently using it to design pan-pathogen panels for other applications). Ultimately, the probe panel for the bat CoV study was limited by budget. Still, the reviewers have raised an important benefit of probe capture, especially in discovery/surveillance applications, so we have added the following text to our discussion (Lines 378 to 380):

“Panels could also be expanded to include other zoonotic viral taxa that circulate in bats like paramyxoviruses and filoviruses, thereby streamlining surveillance programs.”

We have modified the reviewers’ suggestion slightly, mentioning paramyxoviruses and filoviruses instead of influenza viruses. Only two subtypes of influenza A virus have been observed in bats (H17N10 and H18N11), and only a handful of examples have been reported. Furthermore, their status as canonical influenza A viruses is debatable, and their zoonotic risk is predicted to be negligible (Brunotte *et al.,* 2016, Ciminski *et al.,* 2019, Ciminski *et al.,* 2020, Mehle *et al.,* 2014, Tong *et al.,* 2012, Tong *et al.,* 2017). Paramyxoviruses and filoviruses, on the other hand, include numerous bat-associated zoonotic pathogens that have caused serious outbreaks, *e.g.* Nipah virus, Hendrah virus, Ebola virus, and Marburg virus.

5. A quantification of the viral load of the samples would be useful to help understand the similarities and differences between the samples post sequencing. For example, SARS-CoV-2 hybrid capture needs a Ct of < 25 to get a reasonable genome whereas ARTIC amplicon sequencing can get similar results at 30. The complexities of working with archival samples and transporting them over long distances limit what can be done.

To address the reviewers’ query, we designed custom RT-qPCR assays to estimate the abundance of viral abundance in these specimens. We designed primers using the previously generated partial RdRp sequences; this provided us with relatively conserved targets that had been characterized in all specimens, including those from which probe capture had not recovered much sequence. Since different partial RdRp amplicons were used to sequence the *α-* and *betacoronaviruses*, we created separate RT-qPCR assays. Consequently, Ct values became incomparable, so we created standard curves and converted all observations to genome copies per μL. There was a strong (r=0.81) and significant (p<0.0001) relationship between viral abundance and extent of genome recovery. These results have been added to Figure 5B and to the text at Lines 230 to 234. We have also updated the legend for Figure 5. We have added text describing the RT-qPCR Materials and methods (Lines 533 to 552, Table 5, and Supplementary file 2).

We have also added Nicole Lerminiaux as co-author (Line 7) for designing the RT-qPCR assays, collecting the RT-qPCR data, and calculating genome copy numbers.

6. From the methods description of the controls and bioinformatics analysis of the sequencing data, there appears to have been cross-contamination during sample preparation for sequencing. There was then an informatics salvage operation on the data. These salvage operations are fraught with danger, particularly where you have very low levels of RNA and are performing assemblies with very low depths of coverage, which is the case in this work. As the original data, as it came off the sequencer, is not available to the public it is not possible for anyone outside of the study to quantify. This would be a QC fail in SARS-CoV-2 sequencing labs and with a repeat from scratch.Were NTCs/blanks used for sequencing at the same time as the rest of the samples?Were there any CoV reads in these NTCs control (before any read scrubbing)?

We would like to clarify that a bioinformatic tool was used to remove index hops and PCR chimeras, not to remove cross-contaminants. This was not done as a “salvage operation” but as a routine component of FASTQ pre-processing, much like trimming adapters, low quality bases, etc. Following probe capture, there is a post-capture PCR that is necessary for: (1) releasing target material from probe oligos, (2) re-constituting the complementary DNA strand of captured material, and (3) amplifying captured material so that there is sufficient mass to perform a second capture or load the sequencer. With challenging viral specimens like these, captured material is low complexity, fragmented, and requires additional cycles of post-capture PCR enrichment. Consequently, the post-capture PCR creates extensive chimeric library molecules that must be removed before analysis. Additionally, since probe capture is performed on pooled libraries, chimera formation also allows library molecules to swap barcodes.

It’s important to point out that this process was not applied to remove cross-contaminants, nor would it be capable of removing cross-contaminants. The bioinformatic tool we used identifies PCR artefacts based on unique molecular indices (UMIs). In brief, random fragmentation of input nucleic acids generates molecules with unique sequence motifs at both ends (we did not use exogenous UMIs in library adapters for extra uniqueness because we expected low abundance of viral material and high CoV diversity between specimens). These sequence motifs are used to identify all read pairs derived from the same template molecule. Index hops are filtered by identifying the library where each read pair is most abundant, then removing instances of those read pairs in other libraries. Similarly, chimeras are filtered by identifying the UMIs that co-occur most frequently, then removing read pairs with mismatched UMIs. True cross-contaminants (*i.e.* RNA molecules introduced from another specimen during library preparation) would have their own UMIs and remain un-filtered by this tool.

The reviewers asked about the use of no-template controls or blanks. As explained in the Materials and methods, study specimen libraries were prepared alongside and from the same master mixes as two control specimens (Lines 491 to 497). Control specimens were composed of human reference RNA spiked with a synthetic control oligo. No bat CoV sequences were observed in these control libraries following chimera removal, so no evidence of cross-contamination was observed in this study. Furthermore, the synthetic control oligo has an artificial, computer-generated sequence. Probes in our panel target this synthetic oligo, allowing us to use it as a positive control. This assures us that our positive control material could not have been a source of CoV cross-contaminants. The absence of control oligo sequences in the bat libraries provided additional evidence that observable cross-contamination had not occurred.

We consider these controls superior to the water NTCs/blanks typically used in many sequencing facilities. We argue that water controls can actually understate the true extent of cross-contamination; by using water, there is insufficient nucleic acid mass to successfully construct a library, thereby failing to reveal low-level cross-contamination. In other words, clean NTC/blank libraries do not necessarily show an absence of cross-contamination, they merely indicate that library prep has failed (except in the most egregious cases where the mass of cross-contaminants is sufficient on its own). By using human reference RNA as background matrix in our negative controls, we have used a more sensitive method for detecting/ruling out cross-contamination.

Incidentally, in our experience, we find probe capture to be less susceptible to cross-contamination than amplicon sequencing. In amplicon sequencing, target enrichment occurs as the first step, generating vast numbers of molecular copies that can serve as cross-contaminants during each subsequent step of library prep. In probe capture, on the other hand, no amplification occurs until after barcodes have been incorporated. This means the concentration of potential cross-contaminants being handled is dozens of orders of magnitude lower. The concentration remains low until barcodes are added, at which point the cross-contamination risk ends because the barcodes allow molecules to be de-multiplexed back to their original library.

Many labs have undertaken SARS-CoV-2 work (sequencing, diagnostics, reagent manufacturers) and there is a widespread low level of background contamination. As an indication of background contamination, how many SARS-CoV-2 reads were present in the read data (before any read scrubbing)?The risk is that the cross-contamination from one overperforming sample can overwhelm an underperforming sample, giving you an erroneous mixed assembly.

We appreciate the reviewers’ concerns that the unprecedented scale of work on SARS-CoV-2 specimens during the COVID-19 pandemic could have created a risk for cross-contamination and artefactual coronavirus genetic sequences in our results. Library construction and probe capture was conducted in an academic research lab separated from SARS-CoV-2 clinical operations. The captured libraries were sequenced on an Illumina platform at a local public health laboratory, but we confirmed that this occurred before that lab had deployed SARS-CoV-2 sequencing for clinical specimens. The deep sequencing was performed at a sequencing core facility specializing in human genomics that was not involved in routine, high-volume sequencing of its province’s SARS-CoV-2 clinical specimens. We also verified that no SARS-CoV-2 sequences were present in our data. We believe this is reflected in the phylogenetic results presented in Figure 9; while this phylogeny was only based on spike gene, it demonstrates that the *betacoronaviruses* recovered from these specimens were quite dissimilar from SARS-CoV-2. We have added a line to manuscript highlighting this finding (Lines 296 to 298).

Other comments:1. A positive benefit of amplicon sequencing that should be highlighted is the ability to detect intrahost viral populations.

The reviewers are correct in pointing out that amplicon sequencing can characterize intrahost viral populations. We would contend, however, that probe capture is even more powerful for this application because it is more conducive to the use of UMIs. Read de-duplication is more accurate with UMIs, allowing for higher accuracy when calculating minor variant allele prevalence. UMIs also assist with chimera removal, providing higher confidence that two alleles are present on the same genome if they occasionally co-occur on the same reads. Regardless, characterizing intrahost viral populations is not a focus of high-throughput discovery/surveillance programs, so we have not discussed it in this manuscript.

2. Please check the version of blast used because the version in the text is quite old (possibly just a copy and paste typo).

We thank the reviewers for pointing this out. Version 2.5.0 was installed as a shared resource on our server, and this is the version we erroneously noted when writing our Materials and methods. The analysis was actually conducted in a dedicated conda environment with BLAST version 2.12.0 installed. We have updated the text to correct this mistake.

3. The assembly sizes are small and marginally larger than amplicon sequencing and one can sporadically get different regions of the genome making comparative analysis challenging. This method should really shine on fresh, high viral load samples, so it would be interesting to see it in action, perhaps in the field with sequencing on a MinION. Some comments on this would be useful.

The reviewers raise the possibility of combining probe capture with MinION sequencing. We believe this alternate sequencing platform offers two potential benefits: first, as the reviewers have pointed out, portability for field work, and second, long reads for describing complete large, captured fragments. These enticing possibilities have been investigated by some of the authors of this manuscript, but they faced some important challenges. Their experience left them skeptical.

The biggest limitation was that MinION library prep methods were not compatible with multiplexed, pooled captures. As explained above, probe capture involves a post-capture PCR, so all library molecules must contain a shared site that can be targeted by PCR primers. Furthermore, all necessary parts of the sequencing adapter must be located between these primer sites, otherwise amplified library molecules would lose them. Primers targeting the Illumina P5/P7 sites work perfectly for this purpose. No equivalent primers are provided for ONT adapters, however, and we could not attempt to design any because the ONT adapter sequences were proprietary. As a work-around, probe capture was conducted on incomplete ONT libraries, specifically after adding the PCR barcoding adapter. This allowed us to use the ONT PCR barcoding primers to perform the post-capture PCR while adding library barcodes. Unfortunately, this also meant that each library had to be captured separately because they were not yet barcoded.

The inability to pool libraries for probe capture made MinION completely impractical for a high-throughput surveillance program, both in terms of cost and labour. Nonetheless, we wanted to continue our evaluation; perhaps the platform’s long read capability and portability could outweigh its singleplex capture limitation in special circumstances (as a reflex assay for certain specimens of high interest, for example). Our capture experiment revealed more limitations, however. First, the lower base calling accuracy hindered the identification of UMIs for index hop and chimera removal, and there was extensive data attrition during pre-processing. Since the output of MinION flowcells is limited to begin with, the amount of valid data generated did not allow for very deep analysis. Granted, this was almost 5 years ago, and ONT has made significant improvements to base calling accuracy since then.

But even if base calling had been perfectly accurate, there was another short-coming: the size distribution of captured reads ended up being comparable to the size distribution of an Illumina library. We attributed this to two possible phenomena. First, the probes may have weaker avidity for longer fragments because of their greater mass. Second, post-capture PCR amplification imposes a size selection towards shorter library molecules (exacerbated by the additional PCR cycles needed for challenging viral specimens). Either way, probe capture on MinION did not effectively leverage the platform’s long read capability.

The bottom line was that we essentially got Illumina-length data, except it was lower quality and much less of it was generated. Furthermore, it took longer to get and cost substantially more because we could not do multiplexed pooled captures. Based on these experiences, we would not advocate this approach to anyone attempting high-throughput genomics for viral discovery/surveillance. We are happy to provide this frank review of our experiences attempting to viral probe capture with MinION sequencing. We also think it is valuable to have these thoughts immortalized in this response. However, we suspect it may be a bit too critical, tangential, and lengthy to incorporate into the main text of the manuscript.

4. The introduction could do a better job of linking back to the literature on the use of hybrid capture for virus sequencing. One paper that comes to mind is https://f1000research.com/articles/4-1062 with the method used for SARS-CoV-2 to great success. There are a lot of papers using hybrid capture for SARS-CoV-2 over the past 2 years demonstrating the relevance of the approach. Please cite the appropriate literature.

We have cited some additional literature about viral probe capture (Line 98), including the reference suggested by the reviewers. We have focused on studies that designed novel, custom probe panels targeting diverse and hypervariable viruses because this is the primary challenge for viral discovery/surveillance applications. Contrary to expectations, we did not find the SARS-CoV-2 capture literature especially relevant or useful for this study. In CoV terms, SARS-CoV-2 is a very small, constrained taxonomic space. Unlike pan-bat CoVs, this makes it a trivial probe design task and limits the broader usefulness of SARS-CoV-2 panels (outside of sequencing SARS-CoV-2 specimens).

That being said, we were hopeful to find and invoke literature showing the successful use of hybridization probes for routine SARS-CoV-2 sequencing during the COVID-19 pandemic; it would have been useful to show that viral probe capture already has an established track record in high-throughput facilities. Unfortunately, we could not find compelling evidence of widespread, large-scale use of probe capture for SARS-CoV-2 sequencing. The SARS-CoV-2 probe capture literature seems limited to research studies (*e.g.* Gerber *et al.,* 2022, Wen *et al.,* 2020, and Nagy-Szakal *et al.,* 2021). Amplicon sequencing, using open-source protocols like ARTIC or commercial kits like COVIDSeq, has been the overwhelming favourite for routine SARS-CoV-2 genomics (including at multiple provincial public health laboratories in Canada where some of the authors of this manuscript work).

Concerns related to how credit has been apportioned to authors, particularly those from DRC.We acknowledge your earlier correspondence, detailing the roles of authors. It is also acknowledged that some aspects of primary research related to these samples have been published in prior work, with appropriate credit provided to the DRC team. That said, this manuscript makes mention of some primary specimens being shipped to Canada (Specific text: "21 unique specimens were shipped to Canada: 15 as RNA extracts only, 2 as unextracted swabs in transport medium, and 4 as both previously extracted RNA and unextracted swabs in transport medium"). The unextracted swabs would classify as primary specimens. I'm sure you appreciate that collection of such material is a complex process and takes much effort, the current study would not have been possible without these primary specimens. Without knowledge of the intimate workings of your research group, the authorship line up where DRC authors who undertook this collection but do not share primary/senior authorship, emerged as a concern.We accept your explanation and note your suggestion to include a more detailed author statement. Please do so. That said, there may be merit in discussing this with your team to determine if the author list best reflects the overall intent of the research, including the partnerships created and what appears to be an excellent collaborative relationship between diverse groups, spanning continents. Considerations of joint (sometimes with more than two authors) primary authorship or senior authorship may best reflect the vibrant and collegial nature of these relationships. As a journal, we cannot be prescriptive, the choice is ultimately up to you and your team but I hope this narrative has provided some guidance.

We recognize the reviewers’ concerns about apportioning credit. We would like to clarify that no primary specimens were collected for this study; all specimens used in this study were re-purposed material remaining from previous studies that had already concluded and been published with the DRC collaborators as first and senior authors. We have amended the relevant section of the Materials and methods to clarify this (Lines 413 to 425) and removed some of the details that had been duplicated from Materials and methods of previous studies. The legend for Table 1 (Lines 849 to 855) has also been updated to clarify this context. We have also added an Author Contributions section (Lines 605 to 613) to explain contributions. We applied the Contributor Roles Taxonomy (CRediT) plus standard attribution practices in biological sciences for ordering the author list.

We also wanted to highlight some of the ways in which our global collaborations are driven by a commitment to strengthening access to science and ensuring that research benefits everyone:

1. Continued scientific collaborations and capacity building. The specimens used here were collected as part of a ten-year intercontinental partnership called the PREDICT program (https://ohi.vetmed.ucdavis.edu/programs-projects/predict-project). To establish pathogen surveillance around the globe, PREDICT trained almost 7,000 people and enhanced over 60 laboratory spaces for infectious disease testing, primarily in low- and middle-income countries (LMICs). The program has ended, and the LMIC members have published extensively as lead and senior authors (https://p2.predict.global/publications), demonstrating LMIC researcher control over samples, testing, and analyses. Despite the end of the 10-year program, members of the PREDICT consortium continue to leverage its networks of collaborators and its archived specimens to improve infectious diseases surveillance and biological understanding, with the present study being an example of this.

2. Authorships for all contributors. We ensured that all contributions to this study were recognized with authorships. Collaborators based in DRC provided remaining specimen material from previous studies that had concluded and been published with LMIC researchers as first and senior authors. The manuscript under review represents a follow-up study in which Canadian authors conducted all probe design, laboratory work, data analysis, interpretation, and manuscript writing. The primary goal of the research remains to improve infectious disease surveillance and diagnostics across the globe.

3. No commercialization. We deliberately decided not to commercialize the probe panel. Designing the probe panel relied on free access to pathogen genome sequences in public databases like GenBank, and thus on the communal work of the global scientific community. In the same spirit, we wanted the intellectual products of this study to be freely available. We deposited the full CoV genomes recovered from this study in GenBank. We deliberately sought to publish the probe panel in an open access journal as a free alternative to commercial viral capture panels. Likewise, the software used to design this panel was previously published in an open access journal so that researchers could design their own panels without relying on commercial services offered by private companies.